



ACCURACY OF SNOW DEPTH ESTIMATION IN MOUNTAIN AND PRAIRIE
ENVIRONMENTS BY AN UNMANNED AERIAL VEHICLE
Phillip Harder[1], Michael Schirmer[1,3], John Pomeroy[1], Warren Helgason[1,2]
[1]Centre for Hydrology, University of Saskatchewan, Saskatoon, Saskatchewan, Canada
[2]Department of Civil and Geological Engineering, University of Saskatchewan, Saskatoon,
Canada
[3]now at WSL Institute for Snow and Avalanche Research SLF, Davos, Switzerland
All correspondence to Phillip Harder: phillip.harder@usask.ca

## ABSTRACT

The quantification of the spatial distribution of snow is crucial to predict and assess snow as a water
resource and understand land-atmosphere interactions in cold regions. Typical remote sensing
approaches to quantify snow depth have focused on terrestrial and airborne laser scanning and recently
airborne (manned and unmanned) photogrammetry. In this study photography from a small unmanned
aerial vehicle (UAV) was used to generate digital surface models (DSMs) and orthomosaics for
snowcovers at a cultivated agricultural Canadian Prairie and a sparsely-vegetated Rocky Mountain alpine
ridgetop site using Structure from Motion (SfM). The ability of this method to quantify snow depth,
changes in depth and its spatial variability was assessed for different terrain types over time. Root mean
square errors in snow depth estimation from the DSMs were 8.8 cm for a short prairie grain stubble
surface, 13.7 cm for a tall prairie grain stubble surface and 8.5 cm for an alpine mountain surface. This
technique provided meaningful information on maximum snow accumulation and snow-covered area
depletion at all sites, while temporal changes in snow depth could also be quantified at the alpine site
due to the deeper snowpack and consequent higher signal-to noise-ratio. The application of SfM to UAV
photographs can estimate snow depth in areas with snow depth > 30 cm – this restricts its utility for
studies of the ablation of shallow, windblown snowpacks. Accuracy varied with surface characteristics,
sunlight and wind speed during the flight, with the most consistent performance found for wind speeds
< 6 m s$^{-1}$, clear skies, high sun angles and surfaces with negligible vegetation cover. Relative to surfaces
having greater contrast and more identifiable features, snow surfaces present unique challenges when
applying SfM to imagery collected by a small UAV for the generation of DSMs. Regardless, the low cost,
deployment mobility and the capability of repeat-on-demand flights that generate DSMs and
orthomosaics of unprecedented spatial resolution provide exciting opportunities to quantify previously
unobservable small-scale variability in snow depth and its dynamics.

## 1. INTRODUCTION

Accumulation, redistribution, sublimation and melt of seasonal or perennial snowcovers are defining
features of cold region environments. The dynamics of snow have incredibly important impacts on land-
atmosphere interactions and can constitute significant proportions of the water resources necessary for
socioeconomic and ecological functions (Armstrong and Brun, 2008; Gray and Male, 1981; Jones et al.,
2001). Snow is generally quantified in terms of its snow water equivalent (SWE) through measurements
of its depth and density. Since density varies less than depth (López-Moreno et al., 2013; Shook and





Gray, 1996) much of the spatial variability of SWE can be described by the spatial variability of snow
depth. Thus, the ability to measure snow depth, and its spatial distribution, is crucial to assess and
predict how the snow water resource responds to meteorological variability and landscape
heterogeneity. Observation and prediction of snow depth spatial distribution is even more relevant with
the anticipated and observed changes occurring due to a changing climate and land use (Dumanski et al.,
2015; Harder et al., 2015; Milly et al., 2008; Mote et al., 2005; Stewart et al., 2004).
The many techniques and sampling strategies employed to quantify snow depth all have strengths and
limitations (Pomeroy and Gray, 1995). Traditionally, manual snow surveys have been used to quantify
snow depth and density along a transect. The main benefit of manual snow surveying is that the
observations are a direct measurement of the snow water equivalent; however, it requires significant
labour, is a destructive sampling method and can be impractical in complex, remote or hazardous terrain
(DeBeer and Pomeroy, 2009; Dingman, 2002). Many sensors exist that can measure detailed snow
properties non-destructively, with a comprehensive review found in Kinar and Pomeroy (2015), but non-
destructive automated sensors, such as acoustic snow depth rangers (Campbell Scientific SR50) or SWE
analyzers (Campbell Scientific CS275 Snow Water Equivalent Sensor), typically only provide point scale
information and may require significant additional infrastructure or maintenance to operate properly.
Remote sensing of snow from satellite and aerial platforms quantify snow extent at large scales. Satellite
platforms can successfully estimate snow-covered area but problems remain in quantifying snow depth,
largely due to the heterogeneity of terrain complexity and vegetation cover. To date, Light Detection And
Ranging (LiDAR) techniques have provided the highest resolution estimates of snow depth spatial
distribution from both terrestrial (Grünewald et al., 2010) and airborne platforms (Hopkinson et al.,
2012). The main limitations encountered are available areas of observation (sensor viewshed) for the
terrestrial scanner and the prohibitive expense and long lead time needed for planning repeat flights for
the aerial scanner (Deems et al., 2013). Typically, airborne LiDAR provides data with a ground sampling
of nearly 1 m and a vertical accuracy of 15 cm (Deems and Painter, 2006; Deems et al., 2013). While
detailed, this resolution still does not provide observations of the spatial variability of snow distributions
that can address microscale processes such as snow-vegetation interactions or wind redistribution in
areas of shallow snowcover, and the frequency of airborne LiDAR observations are typically low, except
for NASA's Airborne Snow Observatory applications in California (Mattmann et al., 2014).
An early deployment of a high resolution digital camera on a remote controlled gasoline powered model
helicopter in 2004 permitted unmanned digital aerial photography to support studies of shrub
emergence and snowcovered area depletion in a Yukon mountain shrub tundra environment (Bewley et
al., 2007). Since then, Unmanned Aerial Vehicles (UAVs) have become increasingly popular for small-scale
high-resolution remote sensing applications in the earth sciences. The current state of the technology is
due to advances in the capabilities and miniaturization of the hardware comprising UAV platforms
(avionics/autopilots, Global-positioning systems (GPS), Inertial Momentum Units (IMUs) and cameras)
and the increases in available computational power to end users for processing imagery. The conversion
of raw images to orthomosaics and digital surface models takes advantage of Structure from Motion
(SfM) algorithms (Westoby et al., 2012). These computationally intensive algorithms simultaneously
resolve camera pose and scene geometry through automatic identification and matching of common
features in multiple images. With the addition of information on the respective camera location, or if
feature locations are known, then georeferenced point clouds, orthomosaics and Digital Surface Models
(DSMs) can be generated (Westoby et al., 2012). Snow is a challenging surface for SfM techniques due to
its relatively uniform surface and high reflectance relative to snow-free areas which limit identifiable



features (Nolan et al., 2015). The resolution of the data products produced by UAVs depends largely on
flight elevation and sensor characteristics but can promise accuracies down to 2.6 cm in the horizontal
and 3.1 cm in the vertical (Roze et al., 2014). The vertical accuracy of the (DSM) is generally 1 - 3 times
the ground sample distance (GSD) (Strecha, 2011). The unprecedented spatial resolution of these
products may be less important than the fact these platforms are deployable at a high, user-defined,
frequency below cloud cover, which can be problematic for airborne or satellite platforms. Manned
aerial platforms have the advantage of covering much larger areas (Nolan et al., 2015) with a more
mature and clear regulatory framework (Marris, 2013; Rango and Laliberte, 2010) than small UAVs.
However, the greater expenses associated with acquisition, maintenance, operation and training of
manned platforms (Marris, 2013), relative to small UAVs, are significant (Westoby et al., 2012). Small
UAVs overcome the limitation of terrestrial LiDAR viewshed constraints and in principle can generate
DSMs equally well for complex and flat terrain. Many snow scientists have expressed great enthusiasm in
the opportunities UAVs present and speculate that the data they produce may drastically change the
quantification of snow accumulation and ablation (Sturm, 2015).
The roots of SfM are found in stereoscopic photogrammetry, which has a long history in topographic
mapping (Collier, 2002). Major advances in the 1990's in computer vision (Boufama et al., 1993;
Spetsakis and Aloimonost, 1991; Szeliski and Kang, 1994) building upon the development of automated
feature matching algorithms (Förstner, 1986; Harris and Step, 1988) has led to the removal of certain
data inputs, such as camera location, orientation or sensor characteristics, which simplifies the
application of this technique. Significant work by the geomorphology community has pushed the
relevance, application and further development of this technique into the earth sciences (Westoby et al.,
2012). Recent application of this technique to snow depth estimation has used imagery captured by
manned aerial platforms (Bühler et al., 2015; Nolan et al., 2015) and increasingly with small UAVs (De
Michele et al., 2015; Vander Jagt et al., 2015; Bühler et al., 2016). These examples have reported vertical
accuracies (root mean square errors) from the manned platforms of 30 cm with horizontal resolution
between 5-20 cm (Nolan et al., 2015) and 2 m (Bühler et al., 2015) and from the UAV 10 cm with a
horizontal of resolution between 50 cm (Vander Jagt et al., 2015) and 10 cm (Bühler et al., 2016). The
accuracy of assessment of the De Michele et al. (2015), Vander Jagt et al. (2015), and Bühler et al. (2016)
studies were limited to a small number of snow depth maps, Bühler et al. (2016) had the most with four
maps, and more are needed to get a complete perspective on the performance of this technique and its
repeatability.
The advent of UAVs and their promise to generate orthomosaics and DSMs of the earth surface at the
centimeter scale at a high observational frequency is exciting. Testing of this technology applied to snow
has been limited, thus a careful assessment is required of the accuracy achievable with varying weather,
terrain, and vegetation, and also of its temporal repeatability. The overall objective of this paper is to
assess the accuracy of snow depth as estimated by imagery collected by small UAVs and processed with
SfM techniques. Specifically, this paper will; 1) assess the accuracy of UAV-derived snow depths with
respect to the deployment conditions and heterogeneity of the earth surface; specifically variability in
terrain relief, vegetation characteristics and snow depth, and 2) identify and assess opportunities for UAV
generated data to advance understanding and prediction of snowcover and snow depth dynamics.



## 2. Sites and Methodology

### 2.1 Sites

The prairie field site (Fig. 1a) is representative of agricultural regions on the cold, windswept Canadian prairies, where agriculture management practices control vegetation physical characteristics which, in turn, influence snow accumulation (Pomeroy and Gray, 1995). There is little elevation relief and the landscape is interspersed with wooded bluffs and wetlands. Snowcover is typically shallow (maximum depth < 50 cm) with development of a patchy and dynamic snow-covered area during melt. Data collection occurred at a field site near Rosthern, Saskatchewan, Canada in spring 2015 as part of a larger project studying the influence of grain stubble exposure on snowmelt processes. The 65-hectare study site was divided into areas of tall stubble (35 cm) and shorter stubble (15 cm). Wheat stubble, clumped in rows ~30 cm apart, remained erect throughout the snow season, which has implications for blowing snow accumulation, melt energetics and snow cover depletion (Fig. 1c). Snow accumulation dynamics and snowmelt energetics in similar environments have been described by Pomeroy et al. (1993, 1998).

The alpine site, located in Fortress Mountain Snow Laboratory in the Canadian Rocky Mountains, is characterized by a ridge oriented in SW-NE direction (Fig. 1b, d) at an elevation of approximately 2300 m. The average slope at the alpine site is ~15 degrees with some slopes > 35 degrees. Large areas of the ridge were kept bare by wind erosion during the winter of 2014/2015 and wind redistribution caused the formation of deep snowdrifts on the leeward (SE) side of the ridge, in surface depressions and downwind of krummholz. Mean snow depth of the snow-covered area at the start of the observation period (May 13, 2015) was 2 m (excluding snow-free areas) with maximum depths over 5 m. The snow albedo differed between clean snow and that which had dust deposition from localized sources. The study area was divided between a North and a South area (red polygons) due to UAV battery and hence flight area limitations. Snow accumulation dynamics and snowmelt energetics in in the same environment have been described by DeBeer and Pomeroy (2010, 2009), MacDonald et al. (2010) and Musselman et al. (2015) and in similar environments by Egli et al. (2012), Grünewald et al. (2010), Mittaz et al. (2015) and Reba et al. (2011).

### 2.2 Methodology

*2.2.1 Unmanned Aerial Vehicle - flight planning – operation - data processing*

A Sensefly Ebee Real Time Kinematic (RTK) UAV (Fig. 2a) was used to collect imagery over both sites. It is marketed as a complete system, including the UAV platform and flight control and image processing software, capable of survey grade accuracy without the use of GCPs (Roze et al., 2014). The Ebee is a hand launched, fully autonomous, battery powered delta wing UAV with a wingspan of 96 cm and a weight of ~0.73 kg including payload. Maximum flight time is up to 45 minutes with cruising speeds between 40-90 km h$^{-1}$. A consumer grade camera, a Canon IXUS, captured imagery that was tagged with location and camera orientation information supplied by RTK corrected Global Navigation Satellite System (GNSS) positioning and IMU, respectively. A Leica GS15 base station supplied the RTK corrections to the UAV that resolve image locations to an accuracy of ± 2.5 cm. Bühler et al. (2015) found that snow depth mapping improved with the use of near-infrared (NIR) imagery as the NIR spectrum is sensitive to variations in snow grain size and water content (Dozier and Painter, 2004), which increases the contrast and complexity of the snow surface. A NIR camera, a customized Canon S110, was also flown repeatedly during this campaign (three times at alpine site and 16 times at prairie site) and captured imagery in three bands; green, red and NIR (850 nm) bands. The Ebee was able to fly in all wind conditions



attempted but image quality, location and orientation became inconsistent and/or was missed when
wind speed at flight altitude approached or exceeded 14 m s$^{-1}$.
At the prairie site, flight altitudes were ~100 m with 60% lateral and 75% longitudinal photo overlaps,
which translated into mapping of up to 100 hectares per flight at a resolution of ~3 cm pixel$^{-1}$. Figure 2b
provides a typical flight plan generated by the eMotion flight control software that was used on the
prairie site. The UAV was flown 22 times during the melt period (6 to 30 March 2015) with three more
flights over a snow free surface between 2 and 9 April 2015. A loaner Ebee, from Spatial Technologies,
the Ebee distributor, performed the first 11 flights at the prairie site due to technical issues with the
Ebee RTK. The geotag errors of the non-RTK loaner Ebee were ±5 m (error of GPS Standard Positioning
Service) and therefore required GCPs to generate georeferenced data products.
Default settings for difficult terrain were chosen for the alpine site, these include a lateral overlap of 85%
and a longitudinal overlap of 75%, with a flight altitude of 100 m. Two flights with perpendicular flight
paths covered the south and north part of the alpine study area. To reduce variations in flight altitudes,
flight plans were adjusted to ensure a more consistent flight altitude using a 1 m resolution DEM, derived
from an available airborne LiDAR scan. The UAV was flown 18 times from 15 May to 24 June 2015 with
four flights over bare ground on 24 July 2015.
Postflight Terra 3D 3 (version 3.4.46) was used to process imagery to generate DSMs and orthomosaics.
Though the manufacturer suggested that they are unnecessary with RTK corrected geotags (error of ±2.5
cm), all processing included GCPs (locations highlighted in Fig. 1). At the prairie site, 10 GCPs comprised
of five tarps and five utility poles were distributed throughout the study area. At the alpine site, the
north and south areas had five and six GCPs, respectively comprised of tarps (Fig. 3a) and easily
identifiable rocks (Fig. 3b) spread over the study area.
Processing involved three steps. First, initial processing extracted features common to multiple images,
optimized external and internal camera parameters for each image, and generated a sparse point cloud.
The second step densified the point cloud and the third step generated a georeferenced orthomosaic
and a DSM. Preferred processing options varied between the sites, with the semi global matching
algorithm in the point densification used to minimize erroneous points that were encountered at the
alpine site (see Sect. 3.3). Generated orthomosaics and DSMs had a horizontal resolution of 3.5 cm at the
prairie site and between 3.5 cm and 4.2 cm at the alpine site.
*2.2.2 Ground truth and snow depth data collection*
To assess the accuracy of the generated DSMs and their ability to measure snow depth, detailed
observations of the land surface elevation and snow depth over the course of snowcover ablation were
made. At the prairie site a GNSS survey, utilizing a Leica GS15 as a base station and another GS15 acting
as a RTK corrected rover, measured the location (x, y and z) of 34 snow stakes to an accuracy of ± 2.5 cm
(locations identified in Fig. 1a). Over the melt period, the snow depth was measured with a ruler (error
of ± 1 cm) along snow surveys between and at each of the 34 snow survey stakes. Combining the snow
depths measured by the snow surveys and their corresponding land surface elevations from the GNSS
survey gives snow surface elevation points that can be directly compared to the UAV derived DSM.
At the alpine site, 100 land surface elevations were measured with a GNSS survey to determine the
general quality of the DSMs. Vegetation was negligible at these locations. For most of the flights a GNSS
survey was also performed on the snowcover. To account for the substantial terrain roughness and to



avoid measurement errors in deep alpine snowpacks, the snowcover surface elevation was directly
determined by the GNSS survey and snow depth was measured with five snow depth measurements in a
0.4 m x 0.4 m square at these locations. The average snow depth of these five values was then compared
to the snow depth determined by the UAV. Time constraints and inaccessible steep snow patches limited
the number of snow depth measurements to between three and 20 measurements per flight.
At both the prairie and alpine site, GCP location measurement employed the same GNSS RTK surveying
method. Snow surveys (maximum one per day) and DSMs (multiple per day) are only compared if from
the same days.
*2.2.3 Snow depth estimation*
Snow depth was estimated by subtracting a DSM representing a snow-free period from a DSM
representing a period with snowcover. This assumes that snow ablation is the only cause of change in the
surface elevations between the dates of image capture. The snow-free DSMs corresponded to imagery
collected on 2 April and 24 July for the prairie and alpine sites, respectively.
*2.2.4 Accuracy assessment*
The accuracy of the UAV-derived DSM or snow depth was estimated by calculating the root mean square
error (RSME), mean error (bias) and standard deviation of the error (SD) with respect to the manual
measurements. The RSME quantifies the overall difference between manually measured and UAV
derived values. Bias quantifies the mean magnitude of the over (positive values) or under (negative
values) prediction of the DSM with respect to manual measurements. The SD quantifies the variability of
the error.
*2.2.5 Signal-to-Noise Calculation*
The signal-to-noise ratio (SNR) compares the level of the snow depth signal with respect to the
measurement error to inform when meaningful information is available. The SNR is calculated as the
mean measured snow depth value divided by the standard deviation of the error between the observed
and estimated snow depths. The Rose criterion, commonly applied in image processing literature, is used
to define the threshold SNR where the UAV returns meaningful snow depth information; this is further
described in Rose (1973). The Rose criterion proposes a SNR ≥ 4 for the condition at which the signal is
sufficiently large to avoid mistaking it for a fluctuation in noise.  Ultimately, the acceptable signal to noise
ratio depends upon the user's error tolerance (Rose, 1973).
## 3. Results and Discussion
### 3.1 Absolute surface accuracy
The accuracy of the DSMs is summarized in Figure 4 and Table 1 by presenting the errors for the
individual flights and a summary of all the flights, respectively. The accuracy of the DSMs relative to the
measured surface points are variable due to dynamic conditions at time of photography and the surface
characteristics. This is seen in the RMSE for individual flights varying from 4 cm to 19 cm. Only a few
problematic flights showed larger RMSE of up to 32 cm, which are marked in blue in Figure 4. In general,
the accuracy of the DSMs as represented by the mean RMSEs in Table 1, were comparable between the
prairie short stubble (8.1 cm), alpine-bare (8.1 cm), alpine-snow (7.5 cm) sites and greater over the
prairie tall stubble (11.5 cm). Besides the five (out of 43 flights) problematic flights, which will be
discussed in section 3.3.1, accuracy was relatively consistent over time at all sites. To clarify, the prairie
flights simultaneously sampled the short and tall stubble areas, thus there were only three problematic



flights at the prairie site in addition to the two at the alpine site (Figure 4). The larger error at the tall
stubble is due to snow and vegetation surface interactions. Over the course of melt, the DSM gradually
became more representative of the stubble surface rather than the snow surface, as the snow surface
dropped below the stubble height. This highlights a problem in applying SfM to estimate snowcover, as
the most prominent features, in this case exposed stubble, are preferentially weighted to represent the
surface. The bias, especially for tall stubble, becomes positive resulting in over prediction of the surface,
as the snow surface drops beneath the stubble height. The number of observations on alpine-snow is
limited (Fig. 4) but no obvious differences were detected with respect to the alpine-bare soil
(determined by 100 observations). These results exclude areas affected by erroneous points, as
described in section 3.3.2, which was small compared to the total snow-covered area.
The manufacturer suggests that RTK level accuracy on the camera geotags without the use of GCPs can
produce products with similar accuracy to those generated with standard GPS positioning and the use of
GCPs (Roze et al., 2014). This was assessed with DSMs created with and without GCPs for flights where
the Ebee's camera geotags had RTK-corrected positions with an accuracy of ± 2.5 cm. This amounted to
nine flights at the prairie site and 22 flights at the alpine site. Inclusion of GCPs had little effect on the
standard deviation of error with respect to surface observations, but resulted in a reduction of the mean
absolute error of the bias from 27 cm to 10 cm and from 14 cm to 6 cm at the prairie and alpine site,
respectively.
The generated NIR DSMs had rough surfaces, large biases and gaps due to SfM not being able to resolve
the surface features. Despite possible advantages over visible imagery due to greater snow contrast, it
was not possible to generate reliable results using the images from this customized Canon S110 NIR
camera.
3.2 Snow depth accuracy
The snow depth errors were similar to that of the surface errors with the alpine and short stubble sites
having very similar errors, with mean RMSEs of 8.5 cm and 8.8 cm, but much larger errors over tall
stubble, with mean RMSE of 13.7 cm (Fig. 5 and Table 2). Snow depth errors were larger than the surface
errors as the errors from the snow-free and snow-covered DSMs are additive in the DSM differencing.
The usability of snow depth determined from DSM differencing requires comparison of signal-to-noise.
Signal-to-noise, SNR in Fig. 5, clearly demonstrates that the deep alpine snowpacks have a large signal
relative to noise and provide very useable information on snow depth both at maximum accumulation
and during most of the snowmelt period (SNR >7). In contrast, the shallow snowpack at the prairie site,
despite a similar absolute error to the alpine site, demonstrates decreased ability to retrieve meaningful
snow depth information over the course of snowmelt; the signal became smaller than the noise.
Applying the Rose criterion of a SNR ~4, it is apparent that only the first flight at the short stubble and
the first two flights at the tall stubble provided useful information on the snow depth signal.
The error of the estimated snow depth is correlated to the bias; this is most apparent at the prairie site
where the estimated, shallow, snow depth varies with the bias. With bias correction, the mean snow
depth, as demonstrated in Fig. 6, shows a relatively coherent time evolution for a shallow snow cover.
Differencing of UAV derived DSMs provides meaningful but limited information about snow depth.
Reliable information is limited to the peak accumulation period at the prairie site, which is typical of
shallow, wind redistributed seasonal snowcovers that cover prairie, steppe and tundra in North and
South America, Europe and Asia. This is in contrast to other studies which suggest this technique can be



universally adopted for snow depth mapping despite reporting a RMSE of up to 30 cm (Bühler et al.,
2015; Nolan et al., 2015). Errors of such a magnitude are inappropriate for estimating the depth of
shallow snowcovers.
3.3 Challenges
*3.3.1 UAV Deployment Challenges*
An attractive attribute of UAVs, relative to manned aerial or satellite platforms, is that they allow "on-
demand" responsive data collection. While deployable under most conditions encountered, the
significant variability in the DSM RMSEs is likely due to the environmental factors at time of flight
including wind conditions, sun angle, flight duration, cloud cover and cloud cover variability. In high wind
conditions (>14 m s$^{-1}$) the UAV struggled to maintain its preprogrammed flight path. This resulted in
missed photos and inconsistent density in the generated point clouds. This UAV does not employ a
gimbal to stabilize camera orientation and thus windy conditions also resulted in blurry images from the
unstable platform that deviate from the ideal vertical orientation. The flights for the DSMs with the
greatest RMSEs had the highest wind speeds as measured by the UAV.
As the system relies on a single camera traversing the areas of interest, anything that may cause a
change in the reflectance properties of the surface will complicate post-processing and influence the
overall accuracy. Consistent lightning is important with a preference for clear, high sun conditions to
minimize shadow dynamics. Diffuse lighting during cloudy conditions resulted in little contrast over the
snow surface and large gaps in the point cloud over snow. Three flights under these conditions could not
be used and were not included in the previously shown statistics. Clear conditions and patchy snowcover
led to large numbers of overexposed pixels (see Sect 3.3.2). Low sun angles should be avoided as
orthomosaics from these times are difficult to classify with respect to the large and dynamic surface
shadows present and the relatively limited reflectance range.
*3.3.2 Challenges applying Structure from Motion over snow*
Erroneous points over snow were generated by post-processing with the default settings at the alpine
sites. These points were up to several metres above the actual snow surface and were mainly located at
the edge of snow patches, but also on irregular and steep snow surfaces in the middle of a snow patch.
The worst cases occurred during clear sunny days over south-facing snow patches, where the whole
snow patch was interspersed with these erroneous points. These points are related to the overexposure
of snow pixels in the raw images, which typically occurred during direct sunlight over a small snow-
covered area. A typical image with overexposed snow pixels had bare ground in the centre and small
snow patches on the edges. The Canon IXUS camera automatically adjusts exposure based on centre-
weighted light metering and is not adjustable. Erroneous points could be eliminated with the removal of
overexposed images. However, reducing the number of images in such a large amount caused a larger
bias and gaps in the point cloud, which made this method inappropriate.
The semi-global matching (SGM) option with optimization for 2.5D point clouds proved to be the best
parameter setting within the post-processing software Postflight Terra 3D. Semi-global matching was
employed to improve results on projects with low or uniform texture images, while the optimization for
2.5D removes points from the densified point cloud (SenseFly, 2015). The SGM option removed most of
the erroneous points with best results if processing was limited to individual flights. Including images
from additional perpendicular flights or merging subareas with overlapping images resulted in a rougher
surface with more erroneous points. This is likely due to changes in the surface lighting conditions



between flights, which challenges SfM. However, there was no additional bias introduced by the use of
SGM and linear artefacts were visible when compared to default settings. These linear artefacts caused
the standard deviation of the error to increase from 1 cm to 3 cm on bare ground. Areas with remaining
erroneous points where identified and excluded from the presented analysis. The ability to reduce these
erroneous points with SGM depended on the version of Postflight Terra 3D. Results achieved with
version 3.4.46 were much better than results from the later version 4.0.81. This suggests that future
users should test different versions to achieve optimal results. The "black box" nature of this proprietary
software and small number of adjustable parameters clearly limits the applications of this post-
processing tool for scientific applications.
3.4 Applications
The distributed snow depth maps generated from UAV imagery are of great utility for understanding
snow processes at previously unrealized resolutions, spatial coverages and frequencies. These products
may directly lead to a greater understanding of snow phenomena and/or inform, initialize and validate
distributed models at a high resolution. Figure 7 provides examples of UAV derived distributed snow
depth maps. The identification of snow dune structures, which correspond to in-field observations, is a
qualitative validation that UAV derived DSM differencing does indeed provide reasonable information on
the spatial variability of snow depth. Actual applications will depend upon the surface, snow depth and
other deployment considerations as discussed.
In the prairies, as discussed earlier, it is reasonable to use this technique to measure peak snow
accumulation. Besides providing an estimate of the total snow volume, this technique can also inform
snow cover depletion curve estimation and description (Pomeroy et al., 1998). Simple snow cover
depletion models can be parameterized with estimates of the mean and standard deviation of the snow
depth (Essery and Pomeroy, 2004), which otherwise are obtained from snow surveying. For 2015, the
bias corrected peak snow accumulation at the short stubble site had a mean of 28.2 cm and a standard
deviation of 7.2 cm while the tall stubble site had a mean of 38 cm and standard deviation of 6.2 cm.
These values correspond to coefficients of variation of 0.255 and 0.173, at the short and tall stubble sites
respectively, which are similar to previous observations from corresponding landforms/surfaces
(Pomeroy et al., 1998). While not discussed in this paper, the classification of the orthomosaics can
quantify snow-covered area (SCA), providing a validation tool for depletion prediction (Fig. 8a).
Orthomosaics have the same horizontal accuracy and resolution as the DSMs; the vertical errors are
irrelevant as orthomosaics lack a vertical component. Interpretation of snow processes from
orthomosaics is therefore possible regardless of surface characteristics or snow depth.
Applications at the alpine site also include the ability to estimate the spatial distribution of snow depth
change due to ablation (Fig. 8b). To obtain ablation rates, the spatial distribution of snow density is still
needed but it may be estimated with a few point measurements or with parameterizations dependent
upon snow depth (Jonas et al., 2009; Pomeroy and Gray, 1995). In Fig. 8b the mean difference in snow
depth between the two flights was 0.9 m; this gives a SNR of ~11 which is more than sufficient to
confidently assess the spatial variability of melt.
Despite the limitations and deployment considerations discussed, UAVs are capable of providing data at
unprecedented spatial and temporal resolutions that can advance understanding of snow processes. The
most important consideration is whether the anticipated signal-to-noise ratio will allow for direct
estimates of snow depth or snow depth change. This limits the use of this technique to areas with snow
depths or observable changes sufficiently larger than the SD of the error. This analysis established this



threshold, at a minimum, to be ~30 cm. This threshold is equal to four times the mean observed SD
(Rose criterion), but will vary with the application, site and user's error tolerance. Regardless of the
accuracy of the absolute surface values, the relative variability within the DSM may offer fresh insights
into the spatial variability of snow depth and snow surface roughness. Previous work on the statistical
properties of snow depth (Deems et al., 2006; Shook and Gray, 1996) and snow surface roughness
(Fassnacht et al., 2009; Manes et al., 2008) could be extended to consider even finer, centimetre-scale,
variability over large areas.

## 4. Conclusions

A new tool, a small UAV that took photographs from which DSMs and orthomosaics were generated
through application of SfM techniques, was evaluated in two different environments, mountain and
prairie, to verify its ability to quantify snow depth and its spatial variability for varying weather
conditions over the ablation period. The introduction of functional UAVs to the scientific community
requires a critical assessment of what can reasonably be expected from these devices over the seasonal
snowcover. Snow represents one of the more challenging surfaces for UAVs and SfM techniques to
resolve due to the lack of contrast and high surface reflectance. Field campaigns assessed the accuracy of
the Ebee RTK system over flat prairie and complex terrain alpine sites subject to wind redistribution and
spatially variable ablation associated with varying surface vegetation and terrain characteristics. The
mean accuracies of the DSMs were 8.1 cm for the short stubble surface, 11.5 cm for the tall surface and
8.7 cm for the alpine site. These DSM errors translate into mean snow depth errors of 8.8 cm, 13.7 cm
and 8.5 cm over the short, tall and alpine sites respectively. Ground control points were needed to
achieve this level of accuracy. Error varied with bias, which allowed application of a bias correction to
improve the accuracy of the snow depth estimates, but this required additional surface observations.
The SfM technique provided meaningful information on maximum snow depth at all sites, and snow
depth depletion could also be quantified at the alpine site due to the deeper snowpack and consequent
higher signal-to-noise ratio. These findings demonstrate that SfM can be applied to accurately estimate
snow depth and its spatial variability only in areas with snow depth > 30 cm. This restricts its application
for shallow, windblown snowcovers. Snow depth estimation accuracy varied with wind speed, surface
characteristics and sunlight; the most consistent performance was found for wind speeds < 6m s$^{-1}$,
surfaces with insignificant vegetation cover, clear skies and high sun angles. The ability to generate good
results declined over especially homogenous snow surfaces and southerly aspects in mountain terrain.
Clear sky conditions were favourable for high snow-covered fractions with limited snow surface
brightness contrast. During snowmelt with reduced snow-covered fraction, clear sky conditions caused
overexposure of snow pixels.
The challenges of applying SfM to imagery collected by a small UAV over snow complicate the generation
of DSMs relative to other surfaces with greater contrast and identifiable features. Regardless, the
unprecedented spatial resolution of the DSMs and orthomosaics, low costs and "on-demand"
deployment provide exciting opportunities to quantify previously unobservable small-scale variability in
snow depth that will only improve the ability to quantify snow properties and processes.





## ACKNOWLEDGEMENTS

The authors wish to acknowledge the reliable assistance of Spatial Geomatics, Ltd of Calgary, Alberta who provided strong technical support and access to a dGPS unit, courtesy Dr. Cherie Westbook, that made this research possible. Funding was provided by NSERC Research Tools and Instruments and Discovery grants, the NSERC Changing Cold Regions Network, the NSERC Postgraduate Scholarships-Doctoral Program, the Global Institute for Water Security and the Canada Research Chairs programme. Logistical support from Fortress Mountain Ski Resort, the University of Calgary Biogeoscience Institute and field assistance from May Guan, Angus Duncan, Kevin Shook, Sebastian Krogh and Chris Marsh of the Centre for Hydrology and post-processing support from Chris Marsh are gratefully noted.



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



Table 1: Absolute surface accuracy summary*

| Area | Variable | Mean (cm) | Maximum (cm) | Minimum (cm) |
|---|---|---|---|---|
| Alpine-bare | RMSE | 8.7 | 15 | 4 |
| Alpine-bare | Bias** | 5.6 | 11 | 1 |
| Alpine-bare | SD | 6.2 | 12 | 3 |
| Alpine-snow | RMSE | 7.5 | 14 | 3 |
| Alpine-snow | Bias** | 4.4 | 13 | 1 |
| Alpine-snow | SD | 5.4 | 13 | 3 |
| Short | RMSE | 8.1 | 12.5 | 4.4 |
| Short | Bias** | 4.4 | 11.2 | 0 |
| Short | SD | 6.3 | 9.5 | 3.2 |
| Tall | RMSE | 11.5 | 18.4 | 4.9 |
| Tall | Bias** | 6.6 | 17.5 | 0.3 |
| Tall | SD | 8.4 | 14.2 | 3.1 |

*summary excludes five flights identified to be problematic due to windy conditions
**mean of absolute values
Table 2: Absolute snow depth accuracy summary*

| Area | Variable | Mean (cm) | Maximum (cm) | Minimum (cm) |
|---|---|---|---|---|
| Alpine | RMSE | 8.5 | 14.0 | 3 |
| Alpine | Bias** | 4.1 | 11.0 | 0 |
| Alpine | SD | 7.1 | 12.0 | 3 |
| Short | RMSE | 8.8 | 15.8 | 0 |
| Short | Bias** | 5.4 | 15.2 | 0 |
| Short | SD | 6.1 | 10.3 | 0 |
| Tall | RMSE | 13.7 | 27.2 | 0 |
| Tall | Bias** | 9.8 | 26.4 | 0 |
| Tall | SD | 8.3 | 13.9 | 0 |

*summary excludes four flights identified to be problematic due to windy conditions
** mean of absolute values



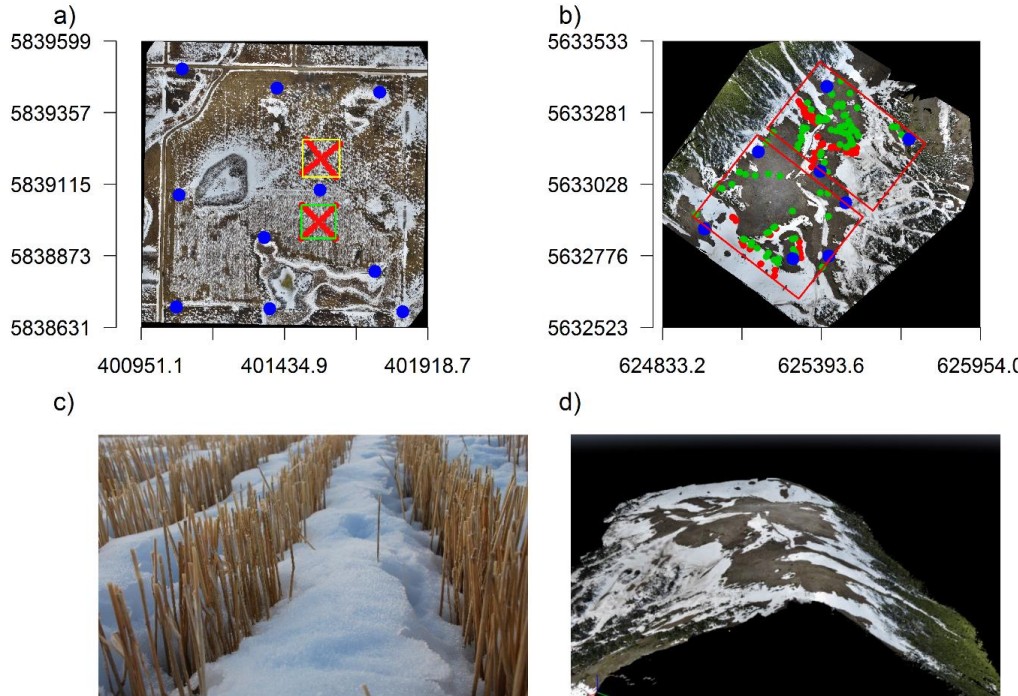


Figure 1: Orthomosaics of a) the prairie site located near Rosthern, Saskatchewan and b) the alpine site at Fortress Mountain Snow Laboratory, Kananaskis, Alberta . The prairie site image (March 19, 2015) has polygons depicting areas used for peak snow depth estimation over short (yellow) and tall (green) stubble. The alpine site image (May 22, 2015) was split into two separately processed subareas (red polygons). Red points in a) and b) are locations of manual snow depth measurements while green points at the alpine site b) were used to test the accuracy of the DSM over the bare surface. Ground control point (GCP) locations are identified as blue points. Axes are UTM coordinates for the prairie site (UTM zone 13N) and alpine site (UTM zone 11N). The defining feature of the prairie site was the c) wheat stubble exposed above the snow surface and at the alpine site was the d) complex terrain as depicted by the generated point cloud (view from NE to SW).





a)                                    b)

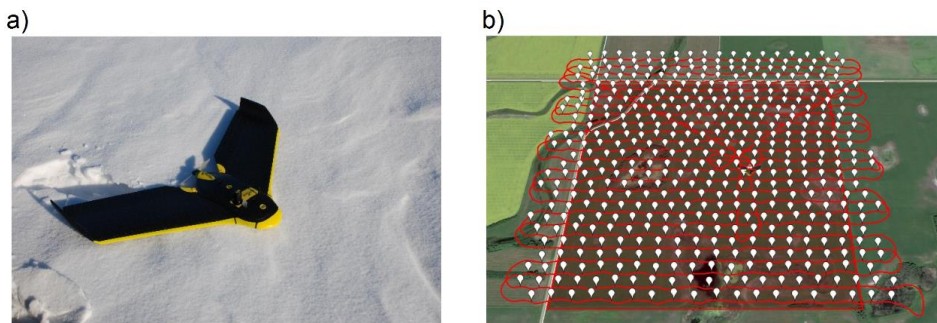

Figure 2: a) Sensefly Ebee RTK, b) a typical flight over the prairie site where red lines represent the flight
path of UAV and the white placemarks represent photo locations.

a)                                    b)

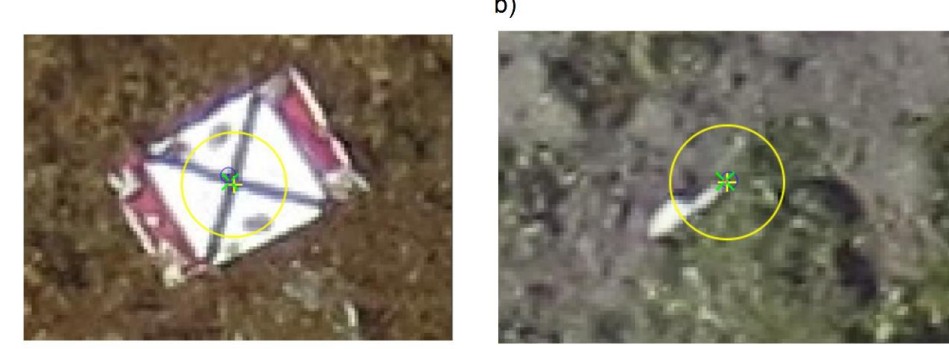

Figure 3: Examples of ground control points that included a) tarps (2.2 m x 1.3 m) and b) identifiable
rocks at the same magnification as the tarp.





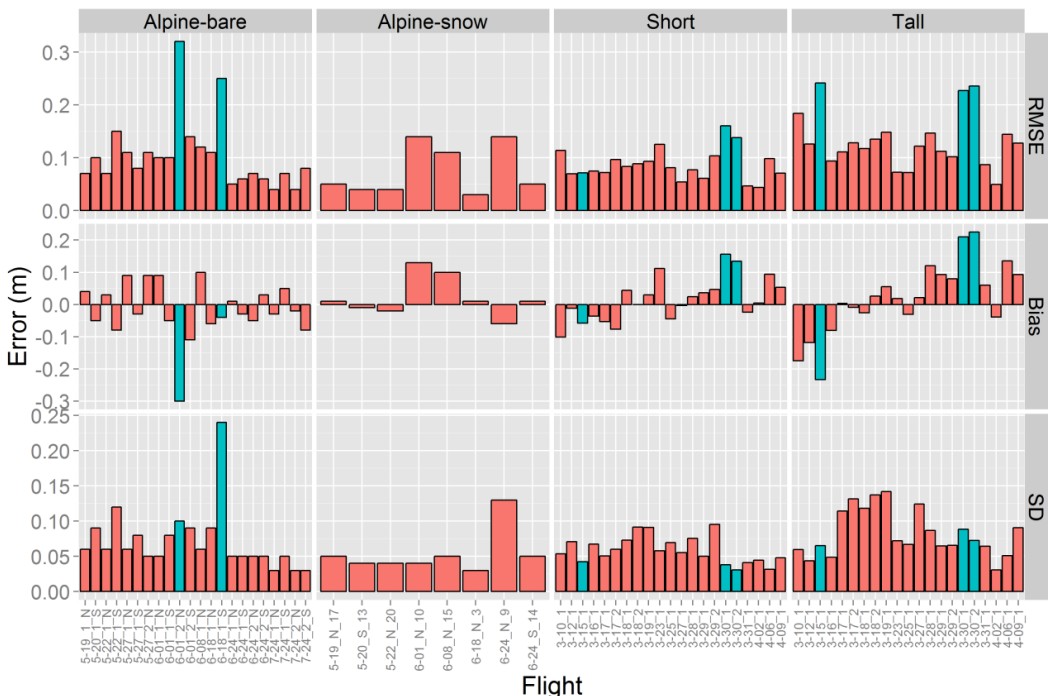

Figure 4: Root mean square error (RMSE, top row), Bias (middle row) and standard deviation (SD) of DSMs with respect to surface over alpine-bare, alpine-snow, and short and tall stubble at prairie site, respectively. Blue bars highlight problematic flights and are excluded from summarization in Table 1. X-axis labels represent month-date-flight number of the day (to separate flights that occurred on the same day). Alpine-bare accuracies are separated into north or south areas, reflected as _N or _S at the end. The last number in the alpine-snow x-axis label is the number of observations used to assess accuracy as they vary between 3 and 20.





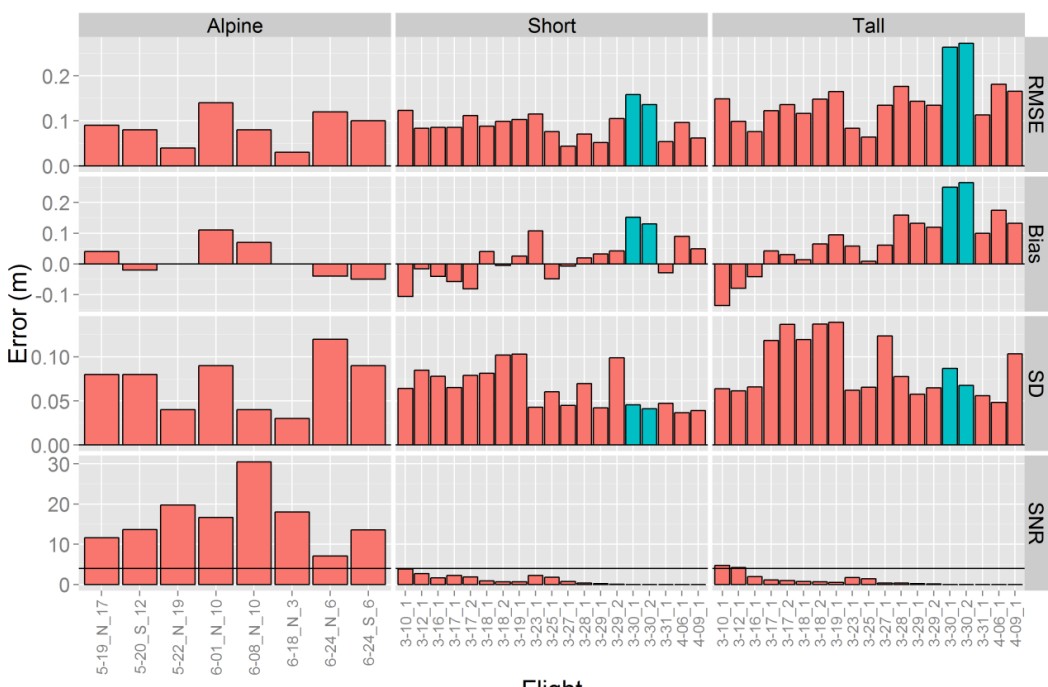

Figure 5: Estimated UAV snow depth error with respect to observed snow depth for short and tall
stubble treatments at prairie site. Blue bars highlight problematic flights and are excluded from
summarization in Table 1. X-axis labels represent month-date. The last value in prairie labels is the flight
of the day (to separate flights that occurred on the same day). Alpine labels separate the north or south
flight areas, reflected as _N or _S respectively, and the last value is the number of observations used to
assess accuracy as they vary between 3 and 19. Horizontal line in the SNR plots is the Rose criterion
(SNR=4) that is used to identify flights with a meaningful snow depth signal.



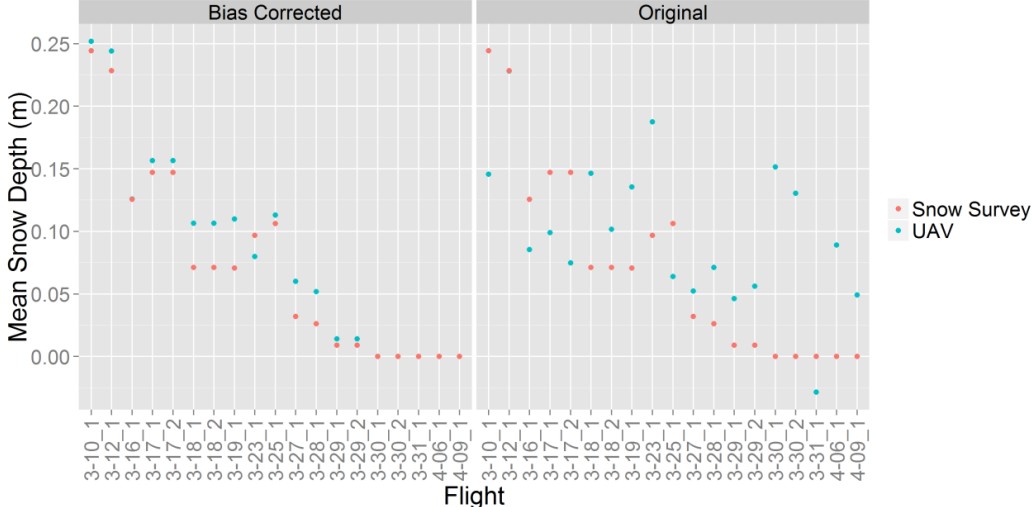


Figure 6: Bias corrected (left column) and original (right column) mean snow depth estimates from the
DSMs (blue points) versus snow survey observations (red points) at the short stubble site. X-axis labels
represent month-date_flight of the day (to separate flights that occurred on the same day).

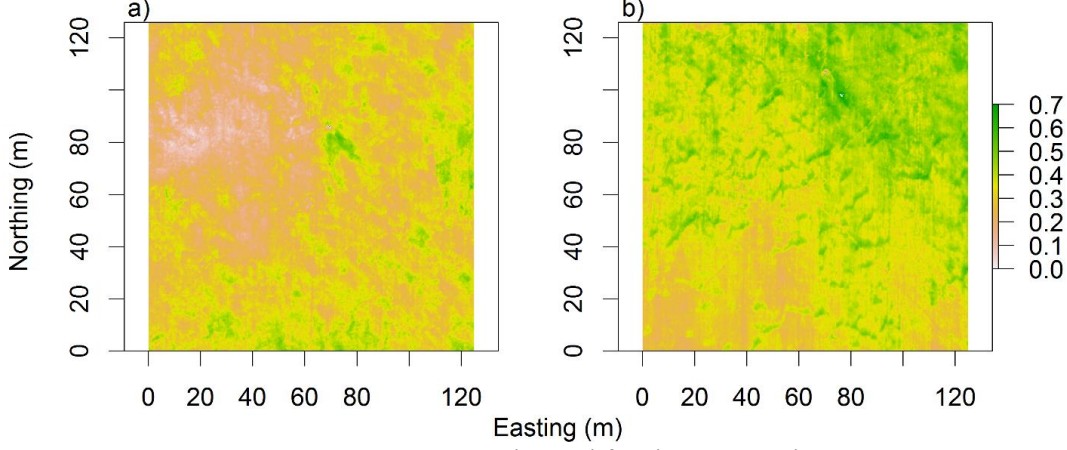

Figure 7: Bias corrected distributed snow depth (meters) for a) short and b) tall stubble treatments at
peak snow depth (March 10, 2015) at the prairie site.





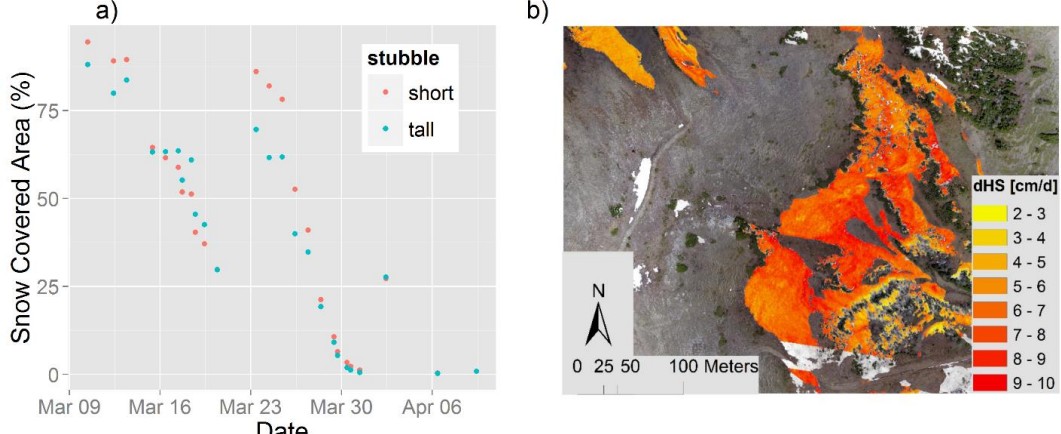

Figure 8: a) Snow covered area depletion over melt for the short and tall stubble sites, with a snowfall
event evident on March 23, and b) snow depth change per day (dHS d$^{-1}$) between May 19 and June 1 in
the northern portion of the alpine site.