# Peer review of "Accuracy of snow depth estimation in mountain and prairie environments by an"

_The Cryosphere, 2016_

## Referee Comment (RC1) · Anonymous Referee #1 · 3 Mar 2016

**Introduction**

This paper analyzes the accuracy of Structure for Motion (SfM) snow depth data products derived from photography acquired with a small unmanned aerial vehicle (UAV). The authors apply a UAV / SfM technique to three distinct environments throughout the spring ablation period. Results from each environment provide quantitative information about snow depth, but based on error analysis (and low signal to noise ratios (SNR)), the authors claim only snowcovers deeper than 30 cm can be reliably estimated with this technique. Despite this limitation, future UAV / SfM technological and operational advances hold promise for observing snowcovers at unprecedented spatial and temporal resolutions.

Developing reliable techniques for collecting snow depth observations at high spatial and temporal resolutions is a line of inquiry directly relevant to The Cryosphere. The authors succeed in presenting novel tools and data. Applications of UAV/ SfM techniques to snow are in their infancy, and the existing literature is sparse and relies on a small number of survey flights. In this case, a large number of survey flights are considered, which is a useful addition to the literature. Using the Rose criterion ( $SNR \geq 4$ ) to assess the quality of snow depth estimates is novel to this study and has not been applied in other recent UAV / SfM studies (Bühler et al., 2016, De Michele et al., 2015, Vander Jagt et al., 2015). The results presented in this study are a valuable contribution to our understanding of what challenges future UAV snow surveys may encounter, especially regarding the influence of vegetation on digital surface models (DSMs) during the ablation period. But the authors are not careful enough in differentiating the limitations of their system from those of other methods of applying SfM to snow, and leave some erroneous

impressions as a result.  With careful attention to correcting that deficiency, as well as an effort to tighten and shorten the paper, it could make a nice contribution once revised.

**General Comments**

More information, context, and discussion regarding the UAV system would help frame the results and conclusions presented in the study. The world of UAVs and their payloads is broad and quickly expanding. Given the diversity of aircraft, cameras, and processing techniques available, the authors should refrain from representing the results from one UAV system (theirs) as indicative of UAV / SfM snow estimation techniques as a whole. Quantitative results may be particular to the UAV system of choice. More discussion of how the choice of aircraft, payload, and processing software may have influenced results is needed.

For example, the Sensefly Ebee Real Time Kinematic aircraft was shown to be sensitive to wind speeds greater than 6 ms$^{-1}$. While this conclusion may be useful to future surveyors (i.e. it may not be worth their time to collect data on windy days), other platforms, such as rotary aircraft or even delta wings with more sophisticated autopilots, may be able to compensate and collect consistent data at higher wind speeds. Do the authors recommend future surveys use rotary platforms? Or does the decreased flight range / endurance of rotary aircraft compared to fixed wings outweigh the increased stability? How much of an operational concern is wind sensitivity, given that snow precipitation events and wind events frequently coincide?

A similar discussion of the camera payload would be useful to readers as well. What is the specific model of the Canon IXUS used in this study? A quick Google search yields at least a dozen different models. What are the specifications of the camera? In particular, what is the bit

depth?  The point about the camera automatically adjusting exposure based on center-weighted values and overexposing some scenes, causing erroneous points, is important. More discussion of this type is useful – for example, that those planning a UAV snow survey should avoid cameras with automatic light metering. Also, the authors mention their system is not equipped with a stabilizing gimbal, which clearly increased wind sensitivity and decreased vertical accuracy. A 3-axis gimbal capable of maintaining an ideal camera orientation is a common feature of many consumer or "prosumer" level UAVs.  A gimbal would certainly increase the quality of the SfM inputs, and therefore perhaps the snow depth resolution. Readers interested in snow, but perhaps UAV/SfM novices, would benefit from a more detailed discussion of the camera system used in the study.

The discussion the authors provide about the Postflight Terra 3D data processing is useful and a good model for how the authors could add context to the aircraft and camera components of the UAV system. "Black box" processing algorithms are a frequent frustration for scientific users trying to understand SfM error sources.

Whether or not the erroneous points caused by overexposure are included in the authors' results is unclear upon first reading. For example, section 3.1 (256 - 257) reads "These results exclude areas affected by erroneous points, as described in section 3.3.2, which was small compared to the total snow-covered area." Which results are the authors referencing? Are the authors speaking generally about every single treatment? Or just the alpine-bare? For example, the authors should consider replacing "These results" with "The alpine-bare results" or "All results." In general, an instance of the word "this" or "these" which lacks a referent can be confusing to the reader because they are unsure as to what precisely the writer is referring. After

reading section 3.1 it seems the authors did not include the erroneous points for some or all of the results - but upon referencing section 3.3.2  (322 - 324) the reader finds conflicting information: "Erroneous points could be eliminated with the removal of overexposed images. However, reducing the number of images in such a large amount caused a larger bias and gaps in the point cloud, which made this method inappropriate." Are the overexposed erroneous points included in the results or not? If the erroneous points are included, specify which results are impacted.

Although the literature is sparse regarding SfM estimates of snow, the authors must be wary of comparing results derived from much different methods. For example, in the discussion section (286 - 292) the results are contrasted against the findings of Nolan et al. 2015, despite their methods using a manned aircraft. Similarly, Buhler et al. 2015 is a reference to a manned aircraft experiment. Given the topic sentence of this section begins "Differencing of *UAV* derived DSMs…" (emphasis mine) some readers may find the contrast of the authors' results with that of a manned aircraft campaign misleading. Also, the 30 cm mean error reported by Nolan et al. is a geolocation error rather than a snow depth error. Snow depth errors were reported as 10 cm, and rigorously documented. Mean snow depths are not reported by Nolan et al. and thus as readers we cannot calculate or assess the SNR of his results, but it does seem like this study is suggesting a higher snow depth threshold for measurements than Nolan et al.  That needs to be addressed.

**Technical Corrections**

**Line 222 - 223:**    "RSME" is twice given as the acronym for root mean square error, rather than RMSE.

**Line 244:** The mean RMSE for the alpine-bare treatment is 8.1 cm, but in Table 1 the value is 8.7 cm.

**Recommendation**

I recommend the authors make revisions to the paper based on the comments above. In general, the authors need to discuss the results appropriately with respect to the referenced work and use more precise language. Also, given the limited scope of this study, readers will prefer a considerably shorter paper. Striving for concision may improve the clarity of the paper as well. The paper could easily be shortened by up about 30%.

---

## Referee Comment (RC2) · Anonymous Referee #2 · 13 Mar 2016

ACCURACY OF SNOW DEPTH ESTIMATION IN MOUNTAIN AND PRAIRIE ENVIRONMENTS BY AN UNMANNED AERIAL VEHICLE

Phillip Harder, Michael Schirmer, John Pomeroy, and Warren Helgason

The authors present an evaluation of snow depth retrieval from airborne photos in challenging snow environments. Rapidly evolving UAV technologies provide a relatively low cost platform to make quick repeat, spatially distributed measurements of snow height. The authors are amongst a handful of people who have attempted to provide a robust evaluation of this technology for retrieval of snow depth. Hence, due to the high

interest in the application of this new measurement technology and the different land surface types presented (prairie and alpine), this paper should be of high value to the community. The detail provided for others wishing to follow their methodology is very useful (such as the niche combination of environmental conditions required to provide optimal accuracy) and the RMSE values as a baseline error estimate are a valuable contribution to the literature. In particular, the use of signal to noise ratios is an excellent addition to the analyses and provides a statistical estimate of error acceptability for this technique. While this is good to see, I would like to raise a number of issues for the authors to consider.

There appear to be a low number (or potentially a low number) of snow depth data used to evaluate depths retrieved from SfM. In some areas of the manuscript this is clear (e.g. observations range between 3 to 19 in the Alpine), but in the prairie, measurements 'between and at 34 snow stakes' is ambiguous. In addition, the reader is left unaware of the spatial coverage of these measurements (within each airborne measurement area) nor how representative they are. At the very least I would expect the n-value to be included in tables 1 and 2. Currently in the literature the amount of in-situ evaluation data for airborne SfM studies are highly variable, e.g. De Michele et al. (2015) tens of depths, Bühler et al. (2015) hundreds of depth, Nolan et al. (2015) thousands of depths. So while this comment should not be seen as an impediment to publication, where very low numbers of in-situ data exist, this needs strong justification or perhaps judicious exclusion from analyses.

Quantification of SCA is demonstrated in Fig 8, and only briefly mentioned in section 3.4. The authors mention this is not discussed in this paper. This leads the reader to ask why not? If data are available to do this in a more thorough manner than currently presented, then this analysis would make an exceptionally valuable contribution to the literature, increase the scientific value of this paper and should definitely be included.

Minor comments:

While NIR imagery was attempted, as it is not used in any of the results or discussion I suggest excluding it from this paper.

While written in a very readable style, the manuscript in its current form could be shortened in many areas, losing extraneous text that is not relevant to the main thrust of the argument. This will provide room for select expansion of sections in greater detail that are currently vague. Some suggestions for sections to delete or shorten considerably are: Ln 11-14; Ln 29-32; Ln 93-97; Ln 98-104; Ln 115-118; Ln 146-149; Ln 152- 155; Ln 266-269; Ln 342-345; Ln 408-412.

Could much of the information in Ln 168- 181 be put in a table, making this section much more concise?

Ln 137: Could the size of the areas measured be explicitly mentioned?

Ln 205: Why was vegetation negligible? I'd like more information about the nature of the vegetation here to justify this claim for the creation of DSMs.

Ln 205 – 'most of the flights' – this is vague. How many flights? Did this affect the analyses?

Ln 219 – (linked to previous vegetation comment) While vegetation is said to be negligible I need more convincing that grasses, particularly on 24 July at the Alpine site after 'spring up' once the snow has cleared, would not have any impact on the on the ability to pick the ground surface from photos. I expect this concern can be allayed through local knowledge, but it needs to be made explicitly and clearly here as it has been a big issue in the past at other sites.

Ln 240: Please give more details describing what 'dynamic conditions' and 'surface characteristics' are.

Ln 242: Please define either here or very clearly in 3.3.1 how 'problematic flights' are defined. Currently this is, at best, vague.

Ln 255: Give more explanation on what is meant by 'limited observations' and why this doesn't affect the detection of differences.

Ln 283: No correlation is presented. Do you mean 'related'? If so please change the terminology? If not, please add the statistical correlations.

Ln 325-340: Uncertain that this section on SGM is that useful. Proprietary software (last sentences of this paragraph) is always problematic for scientific understanding, but somewhat unavoidable for much SfM processing. Also, please explain what '2.5D' means.

Ln 376-381: I consider this just speculation. Suggest removal.

Ln 335: 'were' rather than 'where'.

Ln 373-375: Repetitive use of 'This'. Hard to understand what 'this' is referring to. Please re-write this section with increased clarity.

Ln 472: De Michele et al. 2015 is now in TC rather than TCD.

Ln 597 & 601: Is the mean of the absolute values not the same as RMSE? If so, then stick with RMSE as terminology.

Fig 1 c) – Is this short or tall stubble – please specify.

Fig 5 – Opening sentence of caption - introduce 'Alpine' as well as the prairie sites.

Fig 7 – Add '100' on the y-axis of both plots.

---

## Author Comment (AC1) · 29 Apr 2016

Response to Reviewer 1 regarding: "Accuracy of snow depth estimation in mountain and prairie environments by an unmanned aerial vehicle"

By: Phillip Harder, Michael Schirmer, John Pomeroy, and Warren Helgason

We thank the reviewer for their detailed comments and the points that they raise. Our response will address comments each paragraph at time. Our responses are provided in red text.

Regarding General Comments Paragraph 1:
"More information, context, and discussion regarding the UAV system would help frame the results and conclusions presented in the study. The world of UAVs and their payloads is broad and quickly expanding. Given the diversity of aircraft, cameras, and processing techniques available, the authors should refrain from representing the results from one UAV system (theirs) as indicative of UAV / SfM snow estimation techniques as a whole. Quantitative results may be particular to the UAV system of choice. More discussion of how the choice of aircraft, payload, and processing software may have influenced results is needed."

We agree that articulating our results more clearly in terms of the platform that we use (fixed wing Ebee RTK) will help us to frame our results in the context of recent work that have used multirotor platforms and differentiate more clearly the results that come from unmanned and manned platforms. The revised manuscript now reflects this context more clearly. More information is also included on the UAV platform and camera.

Regarding General Comments Paragraph 2:
"For example, the Sensefly Ebee Real Time Kinematic aircraft was shown to be sensitive to wind speeds greater than 6 ms-1. While this conclusion may be useful to future surveyors (i.e. it may not be worth their time to collect data on windy days), other platforms, such as rotary aircraft or even delta wings with more sophisticated autopilots, may be able to compensate and collect consistent data at higher wind speeds."

A minor edit has been made to the wind sensitivity value. Initially a wind speed greater than 6 m s$^{-1}$ was reported to lead to an increase in DSM errors. Re-examination shows that any differences in DSM error with respect to wind speed were not larger for wind speeds up to 10 m s$^{-1}$ and this value is now used in the paper. This value is obviously platform specific.

"Do the authors recommend future surveys use rotary platforms?"

It has been suggested that multirotor UAV's may be more stable and return better data products in windy conditions (Bühler, et al., 2016). However, there have not been any direct comparison studies that the authors are aware of that validate such assertions. A general statement regarding the use of fixed wing vs. multirotor is challenged by the broad range of UAV designs and capabilities on the market. We see that the only clear benefit of using a multirotor platform is that larger, heavier, potentially more sophisticated, sensors can be

carried (which may improve DSM accuracy as our camera's exposure settings were found to generate erroneous points) and landing accuracy is higher. Disadvantages of multirotor UAVs are that flight speeds and areal coverage are more limited than for fixed wing UAVs. We now note in the manuscript that the Ebee RTK returns data at resolutions that are more than sufficient for our purposes (3cm pixel$^{-1}$), can cover much larger areas and has a higher wind resistance (>14 m/s) than many multirotors – this seems to be a clear overall advantage. Landing accuracy (+/- 5 m) was also sufficient to locate a landing location in the complex topography of the alpine site. The more important issue relative to any comparison between platform types is that all UAVs will have limited flight times and results will be compromised if conditions are windy. A direct comparison between fixed wing and multirotor platforms is necessary to determine exactly how snow depth errors of various platforms may respond to variations in wind speed and lighting conditions. Until then, based on this experience and results of other recent studies (Vander Jagt et al., 2015; Bühler et al., 2016; De Michele et al., 2016), the sufficient image quality, reasonably good high-wind stability, suitable launching and landing procedures for alpine and prairie environments that are noted in the revised manuscript, in conjunction with the clear advantages in fixed wing range, may make fixed wing platforms preferable to the multi-rotor UAVs that have been described in the snow literature to date.

"Or does the decreased flight range / endurance of rotary aircraft compared to fixed wings outweigh the increased stability?"

This is platform specific but comparing this experience and results of other recent studies (Vander Jagt et al., 2015; Bühler et al., 2016; De Michele et al., 2016) would suggest that if the reported errors are similar than the increased range/ endurance of fixed wing platforms hold an advantage. That being said one cannot say anything with certainty without a direct side-by-side comparison. The manuscript has been amended with this discussion as noted above.

"How much of an operational concern is wind sensitivity, given that snow precipitation events and wind events frequently coincide?"

The reviewer does raise the concern that snow precipitation and wind events do sometimes coincide but those events should not be of concern as any UAV should not be flying in a snow event and certainly not in a blowing snow storm because limited visible range (Pomeroy and Male, 1988) would make such operations illegal. Regulatory constraints (in Canada and other regions) restrict operations to visual line of sight, which is significantly hampered by snow in the atmosphere. Practically, airborne snow would significantly obscure surface features as seen from the UAV, reducing its ability to resolve the surface with SfM – there is no point in flying.

The most important consideration when planning to map snow depth with any UAV should be whether the anticipated signal to noise ratio will allow for direct estimates of snow depth or snow depth change. A discussion of platform type and its role in data quality that reflects these points is now in the revised manuscript.

Regarding General Comments Paragraph 3:

"A similar discussion of the camera payload would be useful to readers as well. What is the specific model of the Canon IXUS used in this study? A quick Google search yields at least a dozen different models. What are the specifications of the camera? In particular, what is the bit depth? The point about the camera automatically adjusting exposure based on center-weighted values and overexposing some scenes, causing erroneous points, is important. More discussion of this type is useful – for example, that those planning a UAV snow survey should avoid cameras with automatic light metering. Also, the authors mention their system is not equipped with a stabilizing gimbal, which clearly increased wind sensitivity and decreased vertical accuracy. A 3-axis gimbal capable of maintaining an ideal camera orientation is a common feature of many consumer or "prosumer" level UAVs. A gimbal would certainly increase the quality of the SfM inputs, and therefore perhaps the snow depth resolution. Readers interested in snow, but perhaps UAV/SfM novices, would benefit from a more detailed discussion of the camera system used in the study."

We concur that more details on the camera system would be beneficial. The camera a Canon PowerShot ELPH 110 HS, (which is the same as a Canon IXUS 125 HS) is used to capture red, green and blue band imagery and is modified to be triggered by the autopilot. Exposure settings are automatically adjusted based on a centre-weighted light metering and results may be improved in the future if one could manually adjust exposure settings (not possible with Canon ELPH). Most small fixed wing UAV's do not employ a gimbal due to the space and weight requirements for such arrangements and in the case of the Ebee RTK the camera is fixed in the UAV body. To stabilize the camera when taking photos the UAV cuts power to the motor to minimize vibrations and levels the entire UAV resulting in consistent nadir image orientation. The camera has a 16.1 Mp 1/2.3-inch CMOS sensor and stores images as JPEGs, resulting in images with 8-bit depth for the three color channels. These details are now in the revised manuscript and addressed in the discussion of errors.

Regarding General Comments Paragraph 5:

"Whether or not the erroneous points caused by overexposure are included in the authors' results is unclear upon first reading. For example, section 3.1 (256 - 257) reads "These results exclude areas affected by erroneous points, as described in section 3.3.2, which was small compared to the total snow-covered area." Which results are the authors referencing? Are the authors speaking generally about every single treatment? Or just the alpine-bare? For example, the authors should consider replacing "These results" with "The alpine-bare results" or "All results." In general, an instance of the word "this" or "these" which lacks a referent can be confusing to the reader because they are unsure as to what precisely the writer is referring. After reading section 3.1 it seems the authors did not include the erroneous points for some or all of the results - but upon referencing section 3.3.2 (322 - 324) the reader finds conflicting information: "Erroneous points could be eliminated with the removal of overexposed images. However, reducing the number of images in such a large amount caused a larger bias and gaps in the point cloud, which made this method inappropriate." Are the overexposed erroneous points included in the results or not? If the erroneous points are included, specify which results are impacted."

We appreciate the reviewer's identification of a confusing discussion on the identification and removal (or not) of erroneous points. This discussion has been simplified and limited to section 3.3.2. Some of the erroneous points encountered in early processing, only on alpine snow, coincide with snow surface measurement locations. On certain days, these errors limited the number of useful surface measurements. Incidentally, the erroneous points are located several metres above the surrounding surface, and thus are obvious and simple to exclude and so it does not make sense to include these in the error statistics.

The areas removed for each flight (as a percentage of the total snow covered area (SCA)) varied between 2% at the beginning of melt when the surface was predominantly snow-covered and 22% near the end of melt when a small number of snow patches persisted. The values of the removed SCA are now noted in the revised manuscript. The point of this discussion was to note how we approached the errors in the hope of helping others who may encounter this issue in the future.

Regarding General Comments Paragraph 6:
"Although the literature is sparse regarding SfM estimates of snow, the authors must be wary of comparing results derived from much different methods. For example, in the discussion section (286 - 292) the results are contrasted against the findings of Nolan et al. 2015, despite their methods using a manned aircraft. Similarly, Buhler et al. 2015 is a reference to a manned aircraft experiment. Given the topic sentence of this section begins "Differencing of *UAV* derived DSMs…" (emphasis mine) some readers may find the contrast of the authors' results with that of a manned aircraft campaign misleading. Also, the 30 cm mean error reported by Nolan et al. is a geolocation error rather than a snow depth error. Snow depth errors were reported as 10 cm, and rigorously documented. Mean snow depths are not reported by Nolan et al. and thus as readers we cannot calculate or assess the SNR of his results, but it does seem like this study is suggesting a higher snow depth threshold for measurements than Nolan et al. That needs to be addressed."

We agree it is important to differentiate that the imagery in this study was collected with a small fixed wing UAV rather than a multirotor or manned aircraft. The main difference between these studies is the collection platform, as application of SfM is fundamentally the same. Different processing software: Agisoft versus Pix4D Mapper versus Postflight Terra (and even between versions of Postflight Terra as we noticed) will give different results but the SfM principles are all the same. The differences in platform will lead to differences in the accuracy of image geotags and orientation, image resolution, bit depth and image overlaps. Regardless, very similar errors are being reported from the many recent studies applying SfM to snow despite the range of platforms and software being employed- this suggests to us that the greatest sources of uncertainty is the SfM procedure, followed by the differences in platform characteristics. The revised manuscript differentiates more clearly between the sources of uncertainty and the platforms used in the referenced studies. The 30cm mean error that we

attribute to snow depth error from Nolan et al. (2015) was an error and are grateful that the reviewer brought it to our attention. It is corrected in the revised manuscript.

Regarding Technical Corrections:
Both errors were typos, we thank the reviewer for noticing them, and they are corrected in the revised manuscript.

Regarding Overall recommendation:
"I recommend the authors make revisions to the paper based on the comments above. In general, the authors need to discuss the results appropriately with respect to the referenced work and use more precise language. Also, given the limited scope of this study, readers will prefer a considerably shorter paper. Striving for concision may improve the clarity of the paper as well. The paper could easily be shortened by up about 30%."

We thank the reviewer for these detailed comments. More precise language will be implemented in the revised manuscript and we will adjust how we reference similar studies to more appropriately reflect their results and the platforms they used in contrast to this study. Efforts were made to be more concise and the revised manuscript is shorter than before and includes recommended details on the UAV platform, camera and more explicit discussion of this work with respect to platform type.

---

## Author Comment (AC2) · 29 Apr 2016

Response to Reviewer 2 regarding: "Accuracy of snow depth estimation in mountain and prairie environments by an unmanned aerial vehicle"

By: Phillip Harder, Michael Schirmer, John Pomeroy, and Warren Helgason

We thank reviewer 2 for the constructive comments and the specific edits that will clarify our findings. Our responses are provided in red text.

Regarding General Comment 1:
"There appear to be a low number (or potentially a low number) of snow depth data used to evaluate depths retrieved from SfM. In some areas of the manuscript this is clear (e.g. observations range between 3 to 19 in the Alpine), but in the prairie, measurements 'between and at 34 snow stakes' is ambiguous. In addition, the reader is left unaware of the spatial coverage of these measurements (within each airborne measurement area) nor how representative they are. At the very least I would expect the n-value to be included in tables 1 and 2. Currently in the literature the amount of in-situ evaluation data for airborne SfM studies are highly variable, e.g. De Michele et al. (2015) tens of depths, Bühler et al. (2015) hundreds of depth, Nolan et al. (2015) thousands of depths. So while this comment should not be seen as an impediment to publication, where very low numbers of in-situ data exist, this needs strong justification or perhaps judicious exclusion from analyses."

We agree that the number of verification points in this analysis is quite variable. Manual snow depth observation protocols were different at the alpine and prairie sites due to the dynamics of the melt processes, and logistics. The locations of the manual snow observations were fixed throughout time at the prairie site. Each stubble treatment zone had 17 observation points identified by a physical stake for a total of 34 points at the prairie site. In contrast, the alpine site did not have a fixed snow course and snow depth measurements were limited by logistics and thus ranged between 3 and 19 sites. While the number of snow measurements is limited and variable at the alpine site, there were 100 surface measurements that were continually snow free which that had very similar errors over the course of the campaign to those of the snow surfaces. Considering the snow covered and non-snow covered surface errors together one can see that despite the limited n of error measurements specific to snow, these were not different from the large sample over bare ground. In contrast to other studies which are limited to assessing accuracy over a single or small number of flights we assessed accuracy over a large number of flights over a season. Therefore, the total number of surface observations available to assess accuracy was high. At the alpine site, absolute snow surface accuracy was assessed at 101 points and snow depth accuracy was assessed at 83 (five probe average at each point corresponds to 415 individually probed depths) points. At the prairie site, absolute snow surface and snow depth accuracy was assessed at the same 646 points. This information is now included in the tables. The locations of the points used to assess snow depth and the alpine bare surfaces are plotted in the site figure (Fig 1ab). The prairie site is very homogenous so evaluation points are quite representative of the study area. The alpine evaluation points are not as representative of the areal variation in snowpacks due to steep and inaccessible slopes

but do reflect the variabilities in snow depth observed. These points are clarified in the manuscript.

Regarding General Comment 2:
"Quantification of SCA is demonstrated in Fig 8, and only briefly mentioned in section 3.4. The authors mention this is not discussed in this paper. This leads the reader to ask why not? If data are available to do this in a more thorough manner than currently presented, then this analysis would make an exceptionally valuable contribution to the literature, increase the scientific value of this paper and should definitely be included."

This is a good comment. The quantification of SCA has been added as an objective of the paper and the manuscript section on quantification of SCA has been expanded. The discussion of orthomosaic accuracy is complementary to that for the DSM so not much text is needed to include this. The additional step needed to assess SCA from orthomosaics is to implement a classification scheme and some options such as traditional supervised/unsupervised classification as well as object-oriented classification are now discussed with a clearer example. Compared to estimating snow depth from DSMs, calculating SCA from an orthomosaic is relatively simple and so is discussed concisely.

Specific Edits:

While NIR imagery was attempted, as it is not used in any of the results or discussion I suggest excluding it from this paper.

- For the sake of brevity and lack of results all references to NIR will be removed.

While written in a very readable style, the manuscript in its current form could be shortened in many areas, losing extraneous text that is not relevant to the main thrust of the argument. This will provide room for select expansion of sections in greater detail that are currently vague. Some suggestions for sections to delete or shorten considerably are: Ln 11-14; Ln 29-32; Ln 93-97; Ln 98-104; Ln 115-118; Ln 146-149; Ln 152- 155; Ln 266-269; Ln 342-345; Ln 408-412. Could much of the information in Ln 168- 181 be put in a table, making this section much more concise?

- Many of the identified sections have been edited to reduce redundancy and/or make more concise.

Ln 137: Could the size of the areas measured be explicitly mentioned?

- The prairie site was 65 hectares but the UAV consistently mapped ~100 hectares (to ensure the area of interest was captured). The alpine site was 24 hectares in size. These areas are listed in the revised manuscript.

Ln 205: Why was vegetation negligible? I'd like more information about the nature of the vegetation here to justify this claim for the creation of DSMs.

- Alpine site vegetation was sparse and where it did exist was limited to short grasses on the ridgetop (<10cm) and shrubs and coniferous trees in deep gullies on the shoulders of the ridge. To avoid potential errors in detecting change associated with vegetation obscuring the snow, springing up as snowpack ablated or growing, accuracy assessment points (the 100 points surveyed) with no vegetation (bare ground or exposed rock) were selected. Other errors, such as offsets or tilts, which are minimized through inclusion of GCPs, had a greater impact on DSM accuracy than vegetation. This is clarified in the revised manuscript.

Ln 205 – 'most of the flights' – this is vague. How many flights? Did this affect the analyses?

- Not all flights throughout the measurement campaign had concurrent snow measurements. Only 8 flights did and this is clarified in the revised manuscript

Ln 219 – (linked to previous vegetation comment) While vegetation is said to be negligible I need more convincing that grasses, particularly on 24 July at the Alpine site after 'spring up' once the snow has cleared, would not have any impact on the on the ability to pick the ground surface from photos. I expect this concern can be allayed through local knowledge, but it needs to be made explicitly and clearly here as it has been a big issue in the past at other sites.

- See answer to previous comment regarding vegetation. These grasses were very sparse.

Ln 240: Please give more details describing what 'dynamic conditions' and 'surface characteristics' are.

- Dynamic conditions reflect changes in lighting due to variability in cloud cover and wind over the course of the flight and surface characteristics reflect changes in vegetation exposure and their shadows. This is clarified in the revised manuscript.

Ln 242: Please define either here or very clearly in 3.3.1 how 'problematic flights' are defined. Currently this is, at best, vague.

- Agreed and fixed. Problematic flights were identified upon on examination of the DSMs - we could easily see that the generated surfaces clearly did not represent the snow surface (rough, with gaps in point clouds). For four of these flights this was due to high wind conditions (> 10 ms-1) and challenging light conditions that were also reflected in quite high RMSE values. One flight at the alpine site had a bias much larger than the other flights. To date we have not been able to come up with a reasonable explanation for this situation beyond the fact that it increases with the inclusion of GCPs. Diagnosis of this error is hampered by the "black box" nature of the software, we cannot examine

intermediate steps to determine where the error originates. The identification of these 'problematic flights' is more rigorously defined in section 3.3.1 of the revised manuscript.

Ln 255: Give more explanation on what is meant by 'limited observations' and why this doesn't affect the detection of differences.

- That sentence was poorly constructed and did not convey what was intended. It is changed in the revised manuscript

Ln 283: No correlation is presented. Do you mean 'related'? If so please change the terminology? If not, please add the statistical correlations.

- For the sake of brevity, the brief discussion of bias correction and the associated figures is now removed.

Ln 325-340: Uncertain that this section on SGM is that useful. Proprietary software (last sentences of this paragraph) is always problematic for scientific understanding, but somewhat unavoidable for much SfM processing. Also, please explain what '2.5D' means.

- The section of SGM is very specific to the processing software that we did use and while important to replicate/understand how we dealt with the erroneous points it is now shortened to be more concise. 2.5D refers to the type of point cloud that is used in the DSM generation. 2.5D point clouds are point clouds that do not have overlapping elements. The best way of conceptualizing this is to consider the figure at the following link: https://support.pix4d.com/hc/en-us/articles/202556289-Difference-between-a-3D-and-a-2-5D-Model#gsc.tab=0. This is clarified in the revised manuscript.

Ln 376-381: I consider this just speculation. Suggest removal.
- Removed in the revised manuscript.

Ln 335: 'were' rather than 'where'.
- Corrected in the revised manuscript.

Ln 373-375: Repetitive use of 'This'. Hard to understand what 'this' is referring to. Please re-write this section with increased clarity.
- Agreed. Section is rewritten.

Ln 472: De Michele et al. 2015 is now in TC rather than TCD.
- Reference is now updated

Ln 597 & 601: Is the mean of the absolute values not the same as RMSE? If so, then stick with RMSE as terminology.

- This is the mean of the bias values from the various flights. Since bias can be negative the absolute of bias values is used to ensure that the magnitudes of the biases are preserved. This should read (is updated in revised manuscript) "mean of absolute bias values". This is different from RMSE, which is the root of the mean squared error.

Fig 1 c) – Is this short or tall stubble – please specify.
- Tall stubble and is now specified in the caption.

Fig 5 – Opening sentence of caption - introduce 'Alpine' as well as the prairie sites.
- Corrected in the revised manuscript.

Fig 7 – Add '100' on the y-axis of both plots.
- Corrected in the revised manuscript.

---

## Author Response (AR3)

Author response regarding re-review of: "Accuracy of snow depth estimation in mountain and prairie environments by an unmanned aerial vehicle"

By: Phillip Harder, Michael Schirmer, John Pomeroy, and Warren Helgason

The technical errors noted in the re-review by reviewer #2 have been addressed and additional typographical changes were made to the revised manuscript to remove errors and improve readability.

In light of reviewer 1 not re-reviewing our revised manuscript we are providing a more detailed response to their suggested edits as requested by the editor. The reviewer comments are in black, our general response to the comments are in red with the specific changes between versions highlighted in blue. The comments of reviewer 1 are difficult to identify specifically in the text as the comments were more general in nature and led to subtle changes throughout the paper as well as addition and rewriting of specific sections. To the best of our ability we will highlight what changes were performed with respect to reviewer 1's suggestions and comments.

Regarding General Comments Paragraph 1:
"More information, context, and discussion regarding the UAV system would help frame the results and conclusions presented in the study. The world of UAVs and their payloads is broad and quickly expanding. Given the diversity of aircraft, cameras, and processing techniques available, the authors should refrain from representing the results from one UAV system (theirs) as indicative of UAV / SfM snow estimation techniques as a whole. Quantitative results may be particular to the UAV system of choice. More discussion of how the choice of aircraft, payload, and processing software may have influenced results is needed."

We agree that articulating our results more clearly in terms of the platform that we use (fixed wing Ebee RTK) will help us to frame our results in the context of recent works that have used multirotor platforms and differentiate more clearly the results that come from unmanned and manned platforms.

In the introduction the results between manned and unmanned systems are more clearly communicated:

Previously (lines 107-114):

These examples have reported vertical accuracies (root mean square errors) from the manned platforms of 30 cm with horizontal resolution between 5-20 cm (Nolan et al., 2015) and 2 m (Bühler et al., 2015) and from the UAV 10 cm with a horizontal of resolution between 50 cm (Vander Jagt et al., 2015) and 10 cm (Bühler et al., 2016). The accuracy of assessment of the De Michele et al. (2015), Vander Jagt et al. (2015), and Bühler et al. (2016) studies were limited to a small number of snow depth maps, Bühler et al. (2016) had the most with four maps, and more

are needed to get a complete perspective on the performance of this technique and its repeatability.

Now (lines 131-139):
The manned aircraft examples have reported vertical accuracies of 10cm (Nolan et al., 2015) and 30 cm (Bühler et al., 2015) with horizontal resolutions of 5-20 cm (Nolan et al., 2015) and 2 m (Bühler et al., 2015). Unmanned aircraft examples have shown similar accuracies and resolution with vertical errors of reported to be ~10 cm with horizontal resolutions between 50 cm (Vander Jagt et al., 2015) and 10 cm (Bühler et al., 2016). The accuracy assessments of the De Michele et al. (2016), Vander Jagt et al. (2015), and Bühler et al. (2016) studies were limited to a small number of snow depth maps. Bühler et al. (2016) had the most with four maps, but more are needed to get a complete perspective on the performance of this technique and its repeatability under variable conditions.

An additional section was added to the discussion to place our results in the context of fixed wing vs. multirotor and manned versus un-manned systems

Added to end of section 3.3.1 (lines 330-344)

It is suggested that multirotor UAVs may be more stable and return better data products in windy conditions (Bühler, et al., 2016). There have not been any direct comparison studies that the authors are aware of that validate such assertions. A general statement regarding the use of fixed wing versus multirotor is also impossible with the broad spectrum of UAVs and their respective capabilities on the market. The only clear benefit of using a multirotor platform is that larger, potentially more sophisticated, sensors can be carried and landing accuracy is greater. That being said, the Ebee RTK returns data at resolutions that are more than sufficient for the purposes of this study (3cm pixel$^{-1}$), can cover much larger areas and has a higher wind resistance (>14 m s$^{-1}$ than many multirotor UAVs. Landing accuracy (±5 m) was also sufficient to locate a landing location in the complex topography of the alpine site. The more important issue relative to any comparison between platform types is that all UAVs will have limited flight times and results are compromised if conditions are windy and light is inconsistent. Until a direct platform comparison study is conducted this experience, and results of other recent studies (Vander Jagt et al., 2015; Bühler et al., 2016; De Michele et al., 2016), suggests that fixed wing platforms, relative to multi-rotor platforms, have similar accuracy and deployment constraints but a clear range advantage.

We also articulate that platform arguments are secondary to conditions at the time of deployment

Previously (lines 370-381)
Despite the limitations and deployment considerations discussed, UAVs are capable of providing data at unprecedented spatial and temporal resolutions that can advance understanding of snow processes. The most important consideration is whether the anticipated

signal-to-noise ratio will allow for direct estimates of snow depth or snow depth change. This limits the use of this technique to areas with snow depths or observable changes sufficiently larger than the SD of the error. This analysis established this threshold, at a minimum, to be ~30 cm. This threshold is equal to four times the mean observed SD (Rose criterion), but will vary with the application, site and user's error tolerance. Regardless of the accuracy of the absolute surface values, the relative variability within the DSM may offer fresh insights into the spatial variability of snow depth and snow surface roughness. Previous work on the statistical properties of snow depth (Deems et al., 2006; Shook and Gray, 1996) and snow surface roughness (Fassnacht et al., 2009; Manes et al., 2008) could be extended to consider even finer, centimetre-scale, variability over large areas.

Now (lines 384-395)

Despite the limitations and deployment considerations discussed, the Ebee RTK was capable of providing accurate data at very high spatial and temporal resolutions. A direct comparison between fixed wing and multirotor platforms is necessary to determine how snow depth errors may respond to variations in wind speed and lighting conditions. Until then, based on this experience and results of other recent studies (Vander Jagt et al., 2015; Bühler et al., 2016; De Michele et al., 2016), we do not expect there to be large differences in errors between platform type. Rather, the most important consideration when planning to map snow depth with a UAV should be whether the anticipated SNR will allow for direct estimates of snow depth or snow depth change. The SNR issue limits the use of this technique to areas with snow depths or observable changes sufficiently larger than the SD of the error. We propose a mean snow depth threshold of 30 cm is necessary to obtain meaningful information on snow depth distribution with current technology. This threshold is equal to four times the mean observed SD (Rose criterion), but will vary with the application, site and user's error tolerance.

More information is also included on the UAV platform and camera.
Previously (lines 152-167):

A Sensefly Ebee Real Time Kinematic (RTK) UAV (Fig. 2a) was used to collect imagery over both sites. It is marketed as a complete system, including the UAV platform and flight control and image processing software, capable of survey grade accuracy without the use of GCPs (Roze et al., 2014). The Ebee is a hand launched, fully autonomous, battery powered delta wing UAV with a wingspan of 96 cm and a weight of ~0.73 kg including payload. Maximum flight time is up to 45 minutes with cruising speeds between 40-90 km h$^{-1}$. A consumer grade camera, a Canon IXUS, captured imagery that was tagged with location and camera orientation information supplied by RTK corrected Global Navigation Satellite System (GNSS) positioning and IMU, respectively. A Leica GS15 base station supplied the RTK corrections to the UAV that resolve image locations to an accuracy of ± 2.5 cm. Bühler et al. (2015) found that snow depth mapping

improved with the use of near-infrared (NIR) imagery as the NIR spectrum is sensitive to variations in snow grain size and water content (Dozier and Painter, 2004), which increases the contrast and complexity of the snow surface. A NIR camera, a customized Canon S110, was also flown repeatedly during this campaign (three times at alpine site and 16 times at prairie site) and captured imagery in three bands; green, red and NIR (850 nm) bands. The Ebee was able to fly in all wind conditions attempted but image quality, location and orientation became inconsistent and/or was missed when wind speed at flight altitude approached or exceeded 14 m s$^{-1}$.

Now (lines 174-190):
A Sensefly Ebee Real Time Kinematic (RTK) UAV (version 01) was used to collect imagery over both sites (Fig. 2a). The platform is bundled with flight control and image processing software to provide a complete system capable of survey grade accuracy without the use of ground control points (GCPs) (Roze et al., 2014). The Ebee RTK is a hand launched, fully autonomous, battery powered, fixed wing UAV with a wingspan of 96 cm and a weight of ~0.73 kg including payload. Maximum flight time is up to 45 minutes with cruising speeds between 40-90 km h$^{-1}$. A modified consumer grade camera, a Canon PowerShot ELPH 110 HS, captures red, green and blue band imagery as triggered by the autopilot. The camera, fixed in the UAV body, lacks a stabilizing gimbal as often seen on multirotor UAVs, and upon image capture levels the entire platform and shuts off motor, to minimize vibration, resulting in consistent nadir image orientation. The camera has a 16.1 MP 1/2.3-inch CMOS sensor and stores images as JPEGs, resulting in images with 8-bit depth for the three color channels. Exposure settings are automatically adjusted based on a center weighted light metering. Images are geotagged with location and camera orientation information supplied by RTK corrected Global Navigation Satellite System (GNSS) positioning and IMU, respectively. A Leica GS15 base station supplied the RTK corrections to the Ebee to resolve image locations to an accuracy of ±2.5 cm. The Ebee was able to fly in all wind conditions attempted but image quality, location and orientation became inconsistent when wind speed at the flight altitude (as observe by an on-board pitot tube) approached 14 m s$^{-1}$.

Regarding General Comments Paragraph 2:
"For example, the Sensefly Ebee Real Time Kinematic aircraft was shown to be sensitive to wind speeds greater than 6 ms-1. While this conclusion may be useful to future surveyors (i.e. it may not be worth their time to collect data on windy days), other platforms, such as rotary aircraft or even delta wings with more sophisticated autopilots, may be able to compensate and collect consistent data at higher wind speeds."

A minor edit has been made to the wind sensitivity value. Initially a wind speed greater than 6 m s$^{-1}$ was reported to lead to an increase in DSM errors. Re-examination shows that any differences in DSM error with respect to wind speed were not larger for wind speeds up to 10 m s$^{-1}$ and this value is now used in the paper. This value is obviously platform specific. No changes to text were made with respect to this comment

"Do the authors recommend future surveys use rotary platforms?"

It has been suggested that multirotor UAV's may be more stable and return better data products in windy conditions (Bühler, et al., 2016). However, there have not been any direct comparison studies that the authors are aware of that validate such assertions. A general statement regarding the use of fixed wing vs. multirotor is challenged by the broad range of UAV designs and capabilities on the market. We see that the only clear benefit of using a multirotor platform is that larger, heavier, potentially more sophisticated, sensors can be carried (which may improve DSM accuracy as our camera's exposure settings were found to generate erroneous points) and landing accuracy is higher. Disadvantages of multirotor UAVs are that flight speeds and areal coverage are more limited than for fixed wing UAVs. We now note in the manuscript that the Ebee RTK returns data at resolutions that are more than sufficient for our purposes (3cm pixel$^{-1}$), can cover much larger areas and has a higher wind resistance (>14 m/s) than many multirotors – this seems to be a clear overall advantage. Landing accuracy (+/- 5 m) was also sufficient to locate a landing location in the complex topography of the alpine site. The more important issue relative to any comparison between platform types is that all UAVs will have limited flight times and results will be compromised if conditions are windy. A direct comparison between fixed wing and multirotor platforms is necessary to determine exactly how snow depth errors of various platforms may respond to variations in wind speed and lighting conditions. Until then, based on this experience and results of other recent studies (Vander Jagt et al., 2015; Bühler et al., 2016; De Michele et al., 2016), the sufficient image quality, reasonably good high-wind stability, suitable launching and landing procedures for alpine and prairie environments that are noted in the revised manuscript, in conjunction with the clear advantages in fixed wing range, may make fixed wing platforms preferable to the multi-rotor UAVs that have been described in the snow literature to date.

This is addressed as detailed in response to previous comments in lines 330-344 and 384-395

"Or does the decreased flight range / endurance of rotary aircraft compared to fixed wings outweigh the increased stability?"

This is platform specific but comparing this experience and results of other recent studies (Vander Jagt et al., 2015; Bühler et al., 2016; De Michele et al., 2016) would suggest that if the reported errors are similar than the increased range/ endurance of fixed wing platforms hold an advantage. That being said one cannot say anything with certainty without a direct side-by-side comparison. The manuscript has been amended with this discussion as noted above. (lines 330-344)

"How much of an operational concern is wind sensitivity, given that snow precipitation events and wind events frequently coincide?"

The reviewer does raise the concern that snow precipitation and wind events do sometimes coincide but those events should not be of concern as any UAV should not be flying in a snow

event and certainly not in a blowing snow storm because limited visible range (Pomeroy and Male, 1988) would make such operations illegal. Regulatory constraints (in Canada and other regions) restrict operations to visual line of sight, which is significantly hampered by snow in the atmosphere. Practically, airborne snow would significantly obscure surface features as seen from the UAV, reducing its ability to resolve the surface with SfM – there is no point in flying.

The most important consideration when planning to map snow depth with any UAV should be whether the anticipated signal to noise ratio will allow for direct estimates of snow depth or snow depth change. A discussion of platform type and its role in data quality that reflects these points is now in the revised manuscript. Lines 384-395

Regarding General Comments Paragraph 3:
"A similar discussion of the camera payload would be useful to readers as well. What is the specific model of the Canon IXUS used in this study? A quick Google search yields at least a dozen different models. What are the specifications of the camera? In particular, what is the bit depth? The point about the camera automatically adjusting exposure based on center-weighted values and overexposing some scenes, causing erroneous points, is important. More discussion of this type is useful – for example, that those planning a UAV snow survey should avoid cameras with automatic light metering. Also, the authors mention their system is not equipped with a stabilizing gimbal, which clearly increased wind sensitivity and decreased vertical accuracy. A 3-axis gimbal capable of maintaining an ideal camera orientation is a common feature of many consumer or "prosumer" level UAVs. A gimbal would certainly increase the quality of the SfM inputs, and therefore perhaps the snow depth resolution. Readers interested in snow, but perhaps UAV/SfM novices, would benefit from a more detailed discussion of the camera system used in the study."

We concur that more details on the camera system would be beneficial. The camera a Canon PowerShot ELPH 110 HS, (which is the same as a Canon IXUS 125 HS) is used to capture red, green and blue band imagery and is modified to be triggered by the autopilot. Exposure settings are automatically adjusted based on a centre-weighted light metering and results may be improved in the future if one could manually adjust exposure settings (not possible with Canon ELPH). Most small fixed wing UAV's do not employ a gimbal due to the space and weight requirements for such arrangements and in the case of the Ebee RTK the camera is fixed in the UAV body. To stabilize the camera when taking photos the UAV cuts power to the motor to minimize vibrations and levels the entire UAV resulting in consistent nadir image orientation. The camera has a 16.1 Mp 1/2.3-inch CMOS sensor and stores images as JPEGs, resulting in images with 8-bit depth for the three color channels. These details are now in the revised manuscript (lines 174-190) and addressed in the discussion of errors.

Previously (lines 295-303)
An attractive attribute of UAVs, relative to manned aerial or satellite platforms, is that they allow "on-demand" responsive data collection. While deployable under most conditions encountered, the significant variability in the DSM RMSEs is likely due to the environmental

factors at time of flight including wind conditions, sun angle, flight duration, cloud cover and cloud cover variability. In high wind conditions (>14 m s$^{-1}$) the UAV struggled to maintain its preprogrammed flight path. This resulted in missed photos and inconsistent density in the generated point clouds. This UAV does not employ a gimbal to stabilize camera orientation and thus windy conditions also resulted in blurry images from the unstable platform that deviate from the ideal vertical orientation. The flights for the DSMs with the greatest RMSEs had the highest wind speeds as measured by the UAV.

Now (lines 309-319)
An attractive attribute of UAVs, versus manned aerial or satellite platforms, is that they allow "on-demand" responsive data collection. While deployable under most conditions encountered, the variability in the DSM RMSEs is likely due to the environmental factors at time of flight including wind conditions, sun angle, flight duration, cloud cover and cloud cover variability. In high wind conditions (>14 m s$^{-1}$) the UAV struggled to maintain its preprogrammed flight path as it was blown off course when cutting power to take photos. This resulted in missed photos and inconsistent density in the generated point clouds. Without a gimballed camera, windy conditions also resulted in images that deviated from the ideal nadir orientation. The flights for the DSMs with the greatest RMSEs had the highest wind speeds as measured by the UAV. Four of the five problematic flights were due to high winds (>10 m s$^{-1}$) and were identified by relatively low-density point clouds with significant gaps which rendered DSMs that did not reflect the snow surface characterises.

Regarding General Comments Paragraph 5:
"Whether or not the erroneous points caused by overexposure are included in the authors' results is unclear upon first reading. For example, section 3.1 (256 - 257) reads "These results exclude areas affected by erroneous points, as described in section 3.3.2, which was small compared to the total snow-covered area." Which results are the authors referencing? Are the authors speaking generally about every single treatment? Or just the alpine-bare? For example, the authors should consider replacing "These results" with "The alpine-bare results" or "All results." In general, an instance of the word "this" or "these" which lacks a referent can be confusing to the reader because they are unsure as to what precisely the writer is referring. After reading section 3.1 it seems the authors did not include the erroneous points for some or all of the results - but upon referencing section 3.3.2 (322 - 324) the reader finds conflicting information: "Erroneous points could be eliminated with the removal of overexposed images. However, reducing the number of images in such a large amount caused a larger bias and gaps in the point cloud, which made this method inappropriate." Are the overexposed erroneous points included in the results or not? If the erroneous points are included, specify which results are impacted."

We appreciate the reviewer's identification of a confusing discussion on the identification and removal (or not) of erroneous points. This discussion has been simplified and limited to section

3.3.2. Some of the erroneous points encountered in early processing, only on alpine snow, coincide with snow surface measurement locations. On certain days, these errors limited the number of useful surface measurements. Incidentally, the erroneous points are located several metres above the surrounding surface, and thus are obvious and simple to exclude and so it does not make sense to include these in the error statistics.

The areas removed for each flight (as a percentage of the total snow covered area (SCA)) varied between 2% at the beginning of melt when the surface was predominantly snow-covered and 22% near the end of melt when a small number of snow patches persisted. The values of the removed SCA are now noted in the revised manuscript (added Table 3-see below).  The point of this discussion was to note how we approached the errors in the hope of helping others who may encounter this issue in the future.

Previously (lines 314-340

[revised manuscript text omitted]

[a]month-day_portion of study area

Regarding General Comments Paragraph 6:
"Although the literature is sparse regarding SfM estimates of snow, the authors must be wary of comparing results derived from much different methods. For example, in the discussion section (286 - 292) the results are contrasted against the findings of Nolan et al. 2015, despite their methods using a manned aircraft. Similarly, Buhler et al. 2015 is a reference to a manned aircraft experiment. Given the topic sentence of this section begins "Differencing of *UAV* derived DSMs…" (emphasis mine) some readers may find the contrast of the authors' results with that of a manned aircraft campaign misleading. Also, the 30 cm mean error reported by Nolan et al. is a geolocation error rather than a snow depth error. Snow depth errors were reported as 10 cm, and rigorously documented. Mean snow depths are not reported by Nolan et al. and thus as readers we cannot calculate or assess the SNR of his results, but it does seem like this study is suggesting a higher snow depth threshold for measurements than Nolan et al. That needs to be addressed."

We agree it is important to differentiate that the imagery in this study was collected with a small fixed wing UAV rather than a multirotor or manned aircraft. The main difference between these studies is the collection platform, as application of SfM is fundamentally the same. Different processing software: Agisoft versus Pix4D Mapper versus Postflight Terra (and even between versions of Postflight Terra as we noticed) will give different results but the SfM principles are all the same. The differences in platform will lead to differences in the accuracy of image geotags and orientation, image resolution, bit depth and image overlaps. Regardless, very similar errors are being reported from the many recent studies applying SfM to snow despite the range of platforms and software being employed- this suggests to us that the greatest sources of uncertainty is the SfM procedure, followed by the differences in platform characteristics. The revised manuscript differentiates more clearly between the sources of uncertainty and the platforms used in the referenced studies (as noted in changes to indtoruction discussed earlier). The 30cm mean error that we attribute to snow depth error from Nolan et al. (2015) was an error and are grateful that the reviewer brought it to our attention. It is corrected in the revised manuscript.

Previously (lines 107-109)
These examples have reported vertical accuracies (root mean square errors) from the manned platforms of 30 cm with horizontal resolution between 5-20 cm (Nolan et al., 2015)
Now (lines 131-132
The manned aircraft examples have reported vertical accuracies of 10cm (Nolan et al., 2015)….

Regarding Technical Corrections:

Both errors were typos, we thank the reviewer for noticing them, and they are corrected in the revised manuscript.

Regarding Overall recommendation:
"I recommend the authors make revisions to the paper based on the comments above. In general, the authors need to discuss the results appropriately with respect to the referenced work and use more precise language. Also, given the limited scope of this study, readers will prefer a considerably shorter paper. Striving for concision may improve the clarity of the paper as well. The paper could easily be shortened by up about 30%."

More precise language was implemented in the revised manuscript and we adjusted how we reference similar studies to more appropriately reflect their results and the platforms they used in contrast to this study (changes to introduction and discussion as already noted). Efforts were made to be more concise (deleting or substantially summarising lines 27-32, 94-95, 100-103, 115-118, 160-165, 176-181, 238-239, 266-269, 283-287, 343-345, 376-381, 395-396). The revised manuscript is a similar length than before and as there is additional information on the UAV platform, camera and more explicit discussion of this work with respect to platform type as requested as well as additions due to reviewer 2 comments.